# *MokA*: Multimodal Low-Rank Adaptation for MLLMs

**Yake Wei[1,2,3] , Yu Miao[1,2,3], Dongzhan Zhou[4], Di Hu[1,2,3]***

[1]Gaoling School of Artificial Intelligence, Renmin University of China
[2]Beijing Key Laboratory of Research on Large Models and Intelligent Governance
[3]Engineering Research Center of Next-Generation Intelligent Search and Recommendation, MOE
[4]Shanghai Artificial Intelligence Laboratory
yakewei@ruc.edu.cn, ymiao@ruc.edu.cn
zhoudongzhan@pjlab.org.cn, dihu@ruc.edu.cn

## Abstract

In this paper, we reveal that most current efficient multimodal fine-tuning methods are hindered by a key limitation: they are directly borrowed from LLMs, often neglecting the intrinsic differences of multimodal scenarios and even affecting the full utilization of all modalities. Inspired by our empirical observation, we argue that unimodal adaptation and cross-modal adaptation are two essential parts for the effective fine-tuning of MLLMs. From this perspective, we propose *Multimodal low-rank Adaptation* (**MokA**), a multimodal-aware efficient fine-tuning strategy that takes multimodal characteristics into consideration. It compresses unimodal information by modality-specific parameters while explicitly enhancing cross-modal interaction, ensuring both unimodal and cross-modal adaptation. Extensive experiments cover three representative multimodal scenarios (audio-visual-text, visual-text, and speech-text), and multiple LLM backbones (LLaMA2/3, Qwen2, Qwen2.5-VL, etc). Consistent improvements indicate the efficacy and versatility of the proposed method. Ablation studies and efficiency evaluation are also conducted to fully asses our method. Overall, we think MokA provides a more targeted solution for efficient adaptation of MLLMs, paving the way for further exploration. The project page is at `https://gewu-lab.github.io/MokA`.

## 1 Introduction

Large language models (LLMs) have gained remarkable popularity due to their impressive ability to understand and generate content. To extend their capabilities to more general multimodal scenarios, recent advancements of Multimodal Large Language Models (MLLMs) [38, 40, 20] have focused on aligning other modalities, such as images, with text tokens, thereby equipping LLMs with the ability to interpret and process content of other modalities. However, due to the massive parameter scale of LLMs, fully fine-tuning such models on downstream tasks is computationally prohibitive and inefficient in most cases.

A promising direction has emerged in the field of LLM fine-tuning before, which involves selectively updating a subset of parameters rather than the full model. These Parameter-Efficient Fine-Tuning (PEFT) strategies have seen widespread adoption and have been successfully extended to the fine-tuning of MLLMs. In particular, LoRA [12] and its variants, which assume that over-parameterized models in fact reside on a low intrinsic dimension, have been broadly applied [6, 7, 39], demonstrating strong adaptability and efficiency. However, the development of efficient multimodal LLM fine-tuning is at present obscured by a "dark cloud": *most current methods are directly borrowed from LLMs, often overlooking the fundamental differences of multimodal scenarios*. Indeed, prior studies

---

*Corresponding Author

39th Conference on Neural Information Processing Systems (NeurIPS 2025).

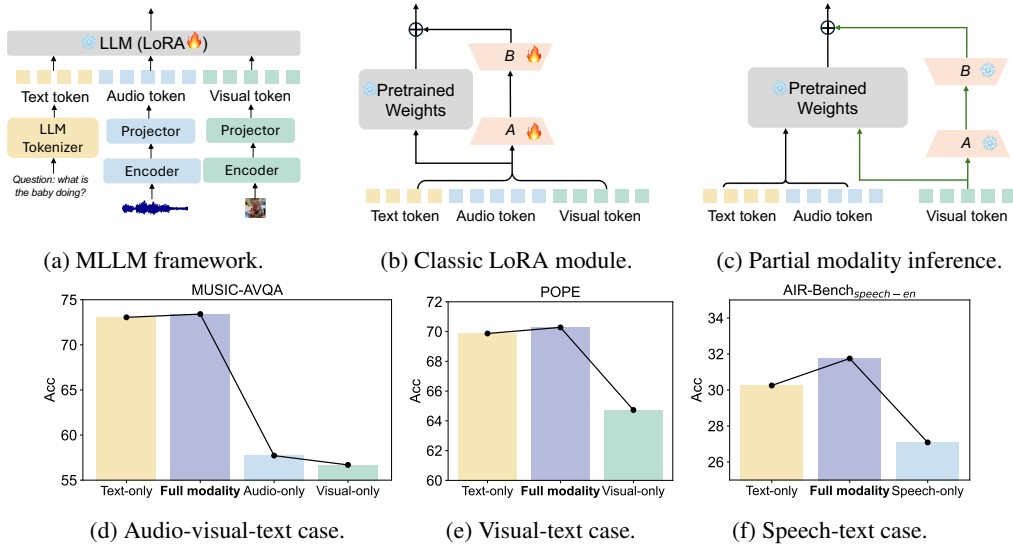

(a) MLLM framework.  (b) Classic LoRA module.  (c) Partial modality inference.

(d) Audio-visual-text case.  (e) Visual-text case.  (f) Speech-text case.

Figure 1: **(a)**: Common MLLM fine-tuning framework. **(b)**: Sketch of classic LoRA module for MLLM fine-tuning. **(c)**: Partial modality inference setting. Take tokens of visual modality as an example. **(d-f)**: Partial modality inference performance of LoRA. *Full modality*: Regular case where all multimodal tokens are processed by the LoRA module. *Text-/Audio-/Visual-/Speech-only*: Only text/audio/visual/speech tokens are passed through the LoRA pathway at the prefilling stage during inference. Results are based on LLaMA2.

in multimodal learning have demonstrated that the inherent heterogeneity of different modalities necessitates modality-specific utilization strategies, rather than a fully unified way [24, 34].

To this end, we are motivated to observe the fine-tuning efficacy of the widely used LoRA strategy. A common MLLM fine-tuning framework is shown in Figure 1a: encoded representation of non-text modality (e.g., audio or visual) is first aligned with the text embedding space via a projector (usually Q-former or MLP), after which the resulting multimodal tokens are integrated and processed jointly by the LLM. In the efficient fine-tuning case, the LLM backbone is frozen, and parameters of additional LoRA modules are optimized. Figure 1b provides the sketch of classic LoRA module. $A$ and $B$ matrices are shared across different modalities.

To further observe how well tokens of different modalities are utilized, we conduct *partial modality inference* experiments. The training stage retains the original setting, wherein all multimodal tokens are processed by the LoRA module. Specifically, we evaluate the model's performance when only tokens from a selected modality are passed through the LoRA adaptation pathway at the prefilling stage during inference. As illustrated in Figure 1c, visual token inference is used as an example. And it should be noted that the pre-trained weights still receive full tokens of all modalities. Results shown in Figure 1d-1f demonstrate a surprising phenomenon across three representative multimodal scenarios, audio-visual-text case, visual-text case, and speech-text case. Text token inference can achieve quite comparable performance to the regular full modalities case. However, Non-text token inference (e.g., audio or visual) leads to a noticeable drop in performance.

The above results suggest that the optimization of all-modality-shared LoRA parameters is overly influenced by text tokens, resulting in non-text tokens being less effectively utilized during fine-tuning. Although these all-modality-shared parameters implicitly improve cross-modal interaction, this phenomenon reveals the need to consider individual modality during fine-tuning. *This fact inspires us that unimodal and cross-modal adaptation are equally critical in the fine-tuning of MLLMs*, which is mostly ignored as mentioned above.

To this end, we propose the **M**ultim**o**dal low-ran**k** **A**daptation (**MokA**), a fine-tuning strategy designed to achieve unimodal adaptation while explicitly enhancing cross-modal interaction. While MokA retains the widely adopted low-rank decomposition matrices, it redefines the roles of matrices $A$ and $B$ to better accommodate multimodal characteristics. Specifically, matrix $A$ is designed to be modality-specific, allowing each modality to compress information independently and thus avoid

interference from others. After that, a cross-attention mechanism is introduced to strengthen the interaction between text tokens and non-text tokens, emphasizing task-relevant features. Finally, a shared multimodal matrix $B$ projects the unimodal low-rank representations into a unified space, facilitating effective alignment across modalities. These three parts jointly ensure both unimodal and cross-modal adaptation. In experiments, noticeable improvement in multiple multimodal scenarios demonstrates the effectiveness of our method. We think MokA represents a first-step attempt at multimodal-aware adaptation, and further possibilities exist under our basis that simultaneously accounts for both unimodal and cross-modal adaptation.

## 2 Method

### 2.1 Rethinking of low-rank adaptation in the multimodal scenario

LoRA [12] is based on the assumption that the weight updates during fine-tuning lie in a subspace of low "intrinsic rank." Rather than updating the entire pre-trained weight matrix directly, LoRA introduces a low-rank decomposition approach, where the update $\Delta W \in \mathbb{R}^{d \times k}$ to a pre-trained matrix $W_0 \in \mathbb{R}^{d \times k}$ is parameterized as the product of two much smaller matrices: $B \in \mathbb{R}^{d \times r}$ and $A \in \mathbb{R}^{r \times k}$, with $r \ll \min(d, k)$. The resulting fine-tuned weight matrix $W'$ is given by $W_0 + \Delta W = W_0 + BA$. Therefore, for $h = W_0 \mathbf{x}$, the modified update forward pass yields:

$$h = W_0 \mathbf{x} + \Delta W \mathbf{x} = W_0 \mathbf{x} + BA \mathbf{x}. \tag{1}$$

Here, $W_0$ remains fixed during training, while only the matrices $A$ and $B$ are learned. To ensure stable training, $A$ is initialized using a uniform Kaiming distribution [11], and $B$ is initialized to zero, leading to an initial update $\Delta W = BA = 0$ at the beginning of fine-tuning. LoRA [12] and its variants have been extensively employed in the parameter-efficient fine-tuning of MLLMs [6, 7, 39]. These methods typically employ shared parameters to uniformly process tokens from all modalities, implicitly facilitating cross-modal interactions during adaptation. However, our empirical results reveal that such shared tuning leads to limited utilization of all modalities. This highlights the need to consider individual modality during fine-tuning.

To better support multimodal adaptation, we argue that both unimodal and cross-modal updates should be considered during fine-tuning. In other words, the model should be able to learn from each modality independently while also ensuring the cross-modal interaction. Therefore, the design of the update mechanism should ensure that both types of information are properly captured during the forward pass:

$$h = W_0 \mathbf{x} + \Delta W \mathbf{x} = W_0 \mathbf{x} + \Delta W [\mathbf{x}^{\mathbf{m}_1}; \mathbf{x}^{\mathbf{m}_2}; \cdots ; \mathbf{x}^{\mathbf{m}_n}], \tag{2}$$

$$= W_0 \mathbf{x} + \underbrace{[\Delta W_1 \mathbf{x}^{\mathbf{m}_1}; \Delta W_2 \mathbf{x}^{\mathbf{m}_2}; \cdots ; \Delta W_n \mathbf{x}^{\mathbf{m}_n}]}_{\text{unimodal adaptation}} + \underbrace{\Delta W_{\mathtt{cross}} [\mathbf{x}^{\mathbf{m}_1}; \mathbf{x}^{\mathbf{m}_2}; \cdots ; \mathbf{x}^{\mathbf{m}_n}]}_{\text{cross-modal adaptation}}, \tag{3}$$

where $n$ is the number of modalities. $\mathbf{x}^{\mathbf{m}_i}$ is the token sequence of modality $i$. $\Delta W_i$ is the unimodal update parameters of modality $i$, and $\Delta W_{\mathtt{cross}}$ is cross-modal update parameters.

### 2.2 Multimodal low-rank Adaptation (MokA)

Based on the above perspective, we propose **M**ultim**o**dal low-ran**k** **A**daptation (**MokA**) strategy, a parameter-efficient fine-tuning method tailored for the multimodal nature of MLLMs. Considering the efficiency advantage of LoRA, MokA retains the core idea of low-rank adaptation, but redefines the roles of the projection matrices $A$ and $B$ to better reflect the characteristics of multimodal scenarios. By unimodal compression and explicitly reinforcing cross-modal interaction, MokA enables both unimodal and cross-modal adaptation, leading to more effective fine-tuning of MLLMs. The overall structure of MokA is depicted in Figure 2.

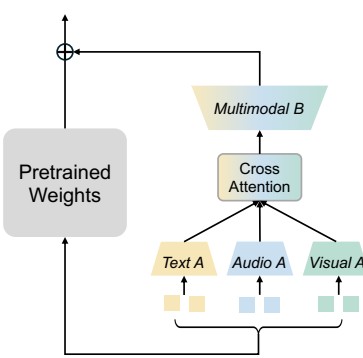

Figure 2: Illustration of MokA strategy.

Concretely, MokA has three core parts: unimodal matrix $A$, task-centric cross-attention, and shared multimodal matrix $B$. Here we take the audio-visual-text case as an example, and other cases can be well extended.

### 2.2.1 Unimodal matrix $A$

For an arbitrary pretrained weight $W_0$ in the LLMs, we suppose its input sequence is $\mathbf{x} = [x_1^a; x_2^a; \cdots; x_{N_a}^a; x_1^v; x_2^v; \cdots; x_{N_v}^v; x_1^t; x_2^t; \cdots; x_{N_t}^t]$. Here $\{N_i\}_{i \in \{a,v,t\}}$ is the token length of modality $i$. $x_1^a$ is the first token of modality $a$, and so on. For simplicity, we use $\mathbf{x}^i$ to denote the token sequence of modality $i$. Then the whole input sequence can be rewritten as $\mathbf{x} = [\mathbf{x}^a; \mathbf{x}^v; \mathbf{x}^t]$.

To ensure the well compression of unique unimodal information and avoid the interruption from others, matrix $A$ is designed to be individual for each modality, allowing tokens from different modalities to be processed independently through their respective parameter. The compressed sequence after matrix $A$ is:

$$A\mathbf{x} = [A^a\mathbf{x}^a; A^v\mathbf{x}^v; A^t\mathbf{x}^t], \tag{4}$$

where $\{A_i\}_{i \in \{a,v,t\}}$ is the parameter of modality $i$. After processing by unimodal matrix $A$, embeddings of each modality are individually mapped into a low-rank space, without the potential influence of other modalities.

### 2.2.2 Task-centric cross-attention

In the fine-tuning process of MLLMs, text and non-text tokens typically serve distinct roles. Specifically, under supervised instruction tuning, text tokens often function as task descriptions or prompts, whereas non-text tokens (e.g., audio or visual inputs) primarily convey contextual information upon which the task is based. The following example illustrates a typical instruction format:

> `<audio>` `<visual>` *Please answer the question: which clarinet makes the sound first?*

In this case, `<audio>` and `<visual>` provide the event information. "*Please answer the question: which clarinet makes the sound first?*" describes the concrete task for LLMs. Successfully answering such questions relies on effectively capturing the semantic association between the task description conveyed by text tokens and the event cues provided by non-text tokens. Therefore, it becomes intuitive and necessary to explicitly emphasize the most relevant cross-modal information to support accurate reasoning. Since unimodal information has been extracted individually after the processing of unimodal matrices $A$, this stage is well-suited for introducing cross-modal interaction. Additionally, as the token embeddings are projected into a low-rank space, the computational burden of performing cross-modal interaction is significantly reduced. Hence, we place the cross-attention part after the low-rank compression to ensure both effectiveness and efficiency. The concrete attention mechanism is illustrated in Figure 3, and is conducted as follows:

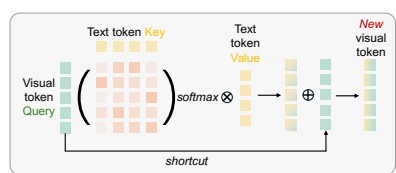

Figure 3: Cross-attention part of MokA. Take the visual token as an example.

$$\texttt{Att}\left(A^a\mathbf{x}^a, A^t\mathbf{x}^t, A^t\mathbf{x}^t\right) = \texttt{softmax}\left(\frac{(A^a\mathbf{x}^a)(A^t\mathbf{x}^t)^\top}{\sqrt{r}}\right) A^t\mathbf{x}^t, \tag{5}$$

$$\texttt{Att}\left(A^v\mathbf{x}^v, A^t\mathbf{x}^t, A^t\mathbf{x}^t\right) = \texttt{softmax}\left(\frac{(A^v\mathbf{x}^v)(A^t\mathbf{x}^t)^\top}{\sqrt{r}}\right) A^t\mathbf{x}^t, \tag{6}$$

where $r$ is the rank. Then, the enhanced audio and visual tokens are:

$$A^a\mathbf{x}^a + \lambda_a\texttt{Att}\left(A^a\mathbf{x}^a, A^t\mathbf{x}^t, A^t\mathbf{x}^t\right), \tag{7}$$

$$A^v\mathbf{x}^v + \lambda_v\texttt{Att}\left(A^v\mathbf{x}^v, A^t\mathbf{x}^t, A^t\mathbf{x}^t\right), \tag{8}$$

where $\lambda_a$ and $\lambda_v$ are the hyperparameters that control the strength of explicit cross-modal interaction. Finally, the sequence after cross-attention is:

$$A\mathbf{x} = [A^a\mathbf{x}^a + \lambda_a\texttt{Att}_{a,t,t}; A^v\mathbf{x}^v + \lambda_v\texttt{Att}_{v,t,t}; A^t\mathbf{x}^t]. \tag{9}$$

Here we use $\texttt{Att}_{i,t,t}$ to simply denote the cross-attention between modality $i$ and text. It should be noted that while we adopt a cross-attention module to explicitly enhance the interaction between text and non-text tokens, alternative designs that serve a similar purpose can also be considered. Further discussion is provided in Section 4.4. In addition, in MokA, linear projections ($W_q$, $W_k$, and $W_v$) are not included in the cross-attention module, since low-rank matrices $A$ of each modality actually can be considered as the linear projection in attention in this case. We also provide more discussion and comparison in Appendix B.

### 2.2.3 Shared multimodal matrix $B$

After unimodal compression and explicit cross-modal interaction enhancement, it becomes crucial to project the resulting unimodal representations into a shared space to facilitate cross-modal alignment. To this end, a shared multimodal matrix $B$ is employed to perform this projection. The final output of the MokA pathway is thus given by:

$$BA\mathbf{x} = [B(A^a\mathbf{x}^a + \lambda_a\mathtt{Att}_{a,t,t}); B(A^v\mathbf{x}^v + \lambda_v\mathtt{Att}_{v,t,t}); BA^t\mathbf{x}^t]. \tag{10}$$

### 2.2.4 Overview

In conclusion, in MokA, for a pretrained weight matrix $W_0 \in \mathbb{R}^{d\times k}$, its update $\Delta W \in \mathbb{R}^{d\times k}$ is parameterized as the product of much smaller matrices: $B \in \mathbb{R}^{d\times r}$ and $\{A^i \in \mathbb{R}^{r\times k}\}_{i\in\{a,v,t\}}$, with $r \ll \min(d,k)$. For input sequence $\mathbf{x}$, the forward pass yields:

$$h = W_0\mathbf{x} + \Delta W\mathbf{x} = W_0\mathbf{x} + \Delta W[\mathbf{x}^a; \mathbf{x}^v; \mathbf{x}^t], \tag{11}$$

$$= W_0\mathbf{x} + [B(A^a\mathbf{x}^a + \lambda_a\mathtt{Att}_{a,t,t}); B(A^v\mathbf{x}^v + \lambda_v\mathtt{Att}_{v,t,t}); BA^t\mathbf{x}^t], \tag{12}$$

$$= W_0\mathbf{x} + \underbrace{[BA^a\mathbf{x}^a; BA^v\mathbf{x}^v; BA^t\mathbf{x}^t]}_{\text{unimodal adaptation}} + \underbrace{[\lambda_a B\mathtt{Att}_{a,t,t}; \lambda_v B\mathtt{Att}_{v,t,t}; \mathbf{0}_{N_t}]}_{\text{cross-modal adaptation}}, \tag{13}$$

where $\mathbf{0}_{N_t}$ denotes the zero vector of dimension $N_t$, since text-token remains unchanged after cross-attention. During fine-tuning, $W_0$ remains unchanged, with $A^i$ and $B$ being subject to optimization. Also, $A^i$ is initialized using the uniform Kaiming distribution [11], while $B$ is initialized to zero. It leads to an initial update $\Delta W = 0$ at the beginning of fine-tuning, to provide a smooth starting point.

Based on Equation 13, MokA ensures both unimodal and cross-modal adaptation, offering a more tailored solution for fine-tuning MLLMs.

## 3 Training and evaluation details

### 3.1 Implement details

Our framework follows the common MLLM framework as illustrated in Figure 1a, but with MokA strategy. Text input is processed by the corresponding LLM tokenizer, and non-text input is first encoded by its encoder, and then aligned with the text embedding space via a projector. Here we use Q-former followed by a two-layer MLP as the projector. Finally, all tokens are fed into LLM.

For the visual branch of audio-visual-text and visual-text scenarios, we use CLIP-ViT/L-14 [25] as the visual encoder to extract the last layer patch level embedding of each frame or image. For the audio branch of the audio-visual-text scenario, we use the BEATs [5] encoder to extract features. For the speech branch of the speech-text scenario, OpenAI's Whisper model [26] is used. The number of query tokens in Q-Former of all branches is 32.

### 3.2 Training procedure and benchmarks

Our experiment of MLLM follows the widely used two-stage training paradigm: pre-training stage that aims to cross-modal alignment and supervised instruction-tuning for downstream tasks.

**Pre-training**: LLM backbone is frozen. Projectors are trainable for cross-modal alignment. For the visual branch of audio-visual-text and visual-text scenarios, trainable modules are trained on video-LLaVA [19] dataset, including the video captioning and the image captioning tasks. For the audio branch of the audio-visual-text scenario, trainable modules are trained on AudioCaps [14] dataset on the audio captioning task. For the speech branch of the speech-text scenario, trainable modules are trained on GigaSpeech-M [4] dataset on the speech recognition task. During pre-training, using the AdamW optimizer with a cosine learning rate schedule. The initial learning rate is $1e-4$ with a warmup ratio of $0.03$.

**Instruction-tuning**: At this stage, we train the model on downstream tasks in different scenarios. Trainable parameters include all projectors and our MokA module. For the audio-visual-text case, the model is fine-tuned on the train set of MUSIC-AVQA [17], and AVE [30], respectively. For the visual-text case, the model is fine-tuned on the LLaVA-Instruct-150K [20] and a 12k subset of

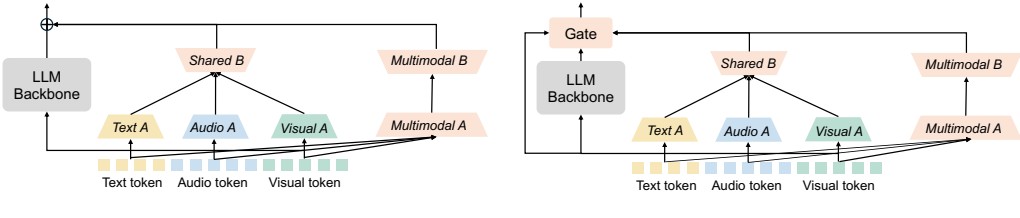

(a) Uni LoRA + MM LoRA.       (b) Uni LoRA + MM LoRA + Gate.

Figure 4: Framework of baselines that also follow our multimodal-aware basis, yet relying on more parameters and offering limited cross-modal interaction.

A-OKVQA. For the speech-text case, the model is fine-tuned on the LibriSpeech [23], using the annotations provided by [28]. Rank of low-rank matrices is 4. The remaining settings are the same as the first stage.

**Inference**: To well assess the effectiveness of our fine-tuning strategy, we evaluate our trained models on in-domain test sets or public benchmarks. Details are provided in the supplementary materials.
- *Audio-visual-text*: in-domain test set of MUSIC-AVQA and AVE dataset.
- *Visual-text*: public benchmarks: $MME_{percep}$ [9], MMBench [22], POPE [18], SEED-Bench [16].
- *Speech-text*: public benchmarks: $MMAU_{mini-speech}$ [27] as well as the foundation subset of $AIR-Bench_{speech-en}$ [37].

**Large Language Model.** For all three cases, LLaMA-2-7b-Chat [31], LLaMA-3-8B-Instruct [10], and Qwen2-7B-Instruct [36] are used as the LLM base model, respectively. For the audio-visual-text case, Qwen2.5-VL-7B-Instruct [2] is also used as the LLM base model. Throughout the training process, weights of LLM are kept frozen. More experiments of Qwen3 are provided in Appendix A.

## 4 Experiments

### 4.1 Audio-visual-text scenario

To validate the effectiveness of our MokA fine-tuning strategy, we compare it with *LoRA* [12] and its variants, including *multiple LoRA*, *LoRAMoE* [8], *DoRA* [21], *HydraLoRA* [29], *Uni-modal LoRA* [1]. In addition, we also compare with two additional baselines, whose frameworks are provided in Figure 4. Concretely, the *Uni LoRA + MM LoRA* strategy employs unimodal low-rank matrices $A$ to extract unimodal information independently, while incorporating an additional fully shared multimodal LoRA module to implicitly promote cross-modal interaction. The *Uni LoRA + MM LoRA + Gate* variant further introduces a gating mechanism to dynamically integrate the outputs of the Uni LoRA and MM LoRA branches for improved fusion. These two baselines incorporate our multimodal-aware basis that ensures both unimodal and cross-modal adaptation, but involve more parameters and offer limited cross-modal interaction. Based on the results in Table 1, we can have the following observations:

Our proposed MokA method achieves the superior overall performance across multiple audio-visual-text datasets, consistently outperforming other baselines and compared methods. While MokA introduces a slight increase in parameter scale compared to standard LoRA, this does not account for the observed performance improvements. Based on the Table 1, multiple LoRA, a baseline that uses 3 A matrices and $B$ A matrices, underperforms both standard LoRA and MokA. Simply increasing the number of low-rank matrices does not necessarily lead to better fine-tuning performance. This suggests that MokA's advantage stems not from parameter quantity, but from its insurance for both unimodal and multimodal adaptation.

The mentioned two baselines, *Uni LoRA + MM LoRA* and *Uni LoRA + MM LoRA* + Gate, achieve competitive results. These results further support the validity of our multimodal-aware basis that unimodal and cross-modal adaptation are both essential for the fine-tuning of MLLMs. However, despite their effectiveness, MokA achieves superior results with fewer parameters and further enhanced cross-modal interactions.

In addition, Qwen2.5VL with LoRA outperforms both LLaMA2 and Qwen2 under the LoRA fine-tuning setting. But when using MokA, the performance of Qwen2.5VL is slightly lower than that of LLaMA2 and Qwen2. A possible reason is that the official visual connector in Qwen2.5VL,

Table 1: Evaluation results of LoRA, its variants, and our MokA on audio-visual-text datasets, MUSIC-AVQA and AVE. #A and #B are the number of low-rank matrices. Here $N = 3$ refers to the number of modalities.

| LLM | Method | MUSIC-AVQA | AVE | #A | #B |
|---|---|---|---|---|---|
| LLaMA2 | LoRA [12] | 73.41 | 69.84 | 1 | 1 |
| | Multiple LoRA | 72.66 | 71.77 | $N$ | $N$ |
| | LoRAMoE [8] | 73.57 | 72.81 | $N$ | $N$ |
| | DoRA [21] | 73.97 | 72.18 | 1 | 1 |
| | HydraLoRA [29] | 74.12 | 72.27 | 1 | $N$ |
| | Uni-modal LoRA [1] | 74.37 | 71.44 | $N$ | $N$ |
| | Uni LoRA + MM LoRA | 74.43 | 72.36 | $N+1$ | 2 |
| | Uni LoRA + MM LoRA + Gate | 74.94 | 73.56 | $N+1$ | 2 |
| | 🍺 MokA | **75.71** | **74.68** | $N$ | 1 |
| Qwen2 | LoRA | 72.83 | 72.13 | 1 | 1 |
| | Multiple LoRA | 72.71 | 72.11 | $N$ | $N$ |
| | LoRAMoE [8] | 73.48 | 72.83 | $N$ | $N$ |
| | DoRA [21] | 73.29 | 72.91 | 1 | 1 |
| | HydraLoRA [29] | 73.14 | 72.59 | 1 | $N$ |
| | Uni-modal LoRA [1] | 73.62 | 73.14 | $N$ | $N$ |
| | Uni LoRA + MM LoRA | 74.09 | 73.35 | $N+1$ | 2 |
| | Uni LoRA + MM LoRA + Gate | 74.71 | 73.96 | $N+1$ | 2 |
| | 🍺 MokA | **75.26** | **74.48** | $N$ | 1 |
| Qwen2.5-VL | LoRA | 73.00 | 71.38 | 1 | 1 |
| | Multiple LoRA | 73.13 | 71.27 | $N$ | $N$ |
| | LoRAMoE [8] | 73.28 | 71.91 | $N$ | $N$ |
| | DoRA [21] | 73.37 | 71.06 | 1 | 1 |
| | HydraLoRA [29] | 73.04 | 71.26 | 1 | $N$ |
| | Uni-modal LoRA [1] | 73.46 | 72.11 | $N$ | $N$ |
| | Uni LoRA + MM LoRA | 73.75 | 72.27 | $N+1$ | 2 |
| | Uni LoRA + MM LoRA + Gate | 73.81 | 72.68 | $N+1$ | 2 |
| | 🍺 MokA | **74.87** | **73.14** | $N$ | 1 |
| LLaMA3 | LoRA | 78.31 | 76.91 | 1 | 1 |
| | Multiple LoRA | 78.63 | 77.02 | $N$ | $N$ |
| | 🍺 MokA | **79.15** | **77.81** | $N$ | 1 |

which serves a similar role to the projector used in our other LLM variants, remains frozen during fine-tuning. As a result, only the newly introduced audio branch is trainable, which may have limited the full potential of our method. But MokA still introduces noticeable improvement in this case, compared to other methods. In summary, our method achieves considerable improvement across various LLM backbones, demonstrating its broad versatility.

## 4.2 Visual-text and speech-text scenarios

To further validate our method across a broader range of multimodal scenarios, we conducted experiments beyond the challenging audio-visual-text case. Specifically, our method is further verified on two representative multimodal scenarios: visual-text and speech-text. For these tasks, we adopted three different LLM backbones, LLaMA2, LLaMA3, and Qwen2. The corresponding results are presented in Table 2 and Table 3. The experimental results demonstrate that our method achieves stable and consistent performance gains across multiple benchmark datasets, further confirming its effectiveness. This indicates the versatility of MokA in handling different multimodal combinations and LLM architectures.

Table 2: Evaluation results of LoRA, its variants, and our MokA on visual-text benchmarks. #A and #B are the number of low-rank matrices. Here $N = 2$ refers to the number of modalities.

| LLM | Method | $MME_{percep}$ | MMBench | POPE | SEED-Bench | #A | #B |
|---|---|---|---|---|---|---|---|
| LLaMA2 | LoRA | 908.52 | 50.64 | 70.28 | 39.71 | 1 | 1 |
| | Multiple LoRA | 882.87 | 49.83 | 68.20 | 38.44 | $N$ | $N$ |
| | LoRAMoE [8] | 938.52 | 51.98 | 71.15 | 39.13 | $N$ | $N$ |
| | DoRA [21] | 786.47 | 51.31 | 71.07 | 38.96 | 1 | 1 |
| | HydraLoRA [29] | 774.47 | 47.33 | 70.87 | 38.81 | 1 | $N$ |
| | Uni-modal LoRA [1] | 992.31 | 51.98 | 72.24 | 39.27 | $N$ | $N$ |
| | Uni LoRA + MM LoRA | 972.87 | 50.96 | 73.39 | 39.74 | $N+1$ | 2 |
| | Uni LoRA + MM LoRA + Gate | 988.37 | 52.01 | 73.48 | 39.91 | $N+1$ | 2 |
| | 🍿 MokA | **1025.86** | **52.74** | **74.23** | **40.45** | $N$ | 1 |
| Qwen2 | LoRA | 1062.34 | 57.89 | 81.17 | 55.25 | 1 | 1 |
| | Multiple LoRA | 1103.28 | 57.01 | 80.96 | 55.13 | $N$ | $N$ |
| | LoRAMoE [8] | 1157.39 | 57.29 | 81.29 | 56.39 | N | N |
| | DoRA [21] | 1024.42 | 56.19 | 80.75 | 55.03 | 1 | 1 |
| | HydraLoRA [29] | 1098.25 | 56.42 | 81.34 | 54.67 | 1 | $N$ |
| | Uni-modal LoRA [1] | 1189.47 | 57.39 | 81.12 | 56.21 | $N$ | $N$ |
| | Uni LoRA + MM LoRA | 1191.81 | 57.17 | 81.46 | 56.84 | $N+1$ | 2 |
| | Uni LoRA + MM LoRA + Gate | 1201.49 | 57.91 | 81.72 | 57.18 | $N+1$ | 2 |
| | 🍿 MokA | **1292.37** | **59.06** | **82.33** | **58.10** | $N$ | 1 |
| LLaMA3 | LoRA | 1030.64 | 68.45 | 77.47 | 56.34 | 1 | 1 |
| | Multiple LoRA | 1032.74 | 68.79 | 78.73 | 56.06 | $N$ | $N$ |
| | 🍿 MokA | **1072.67** | **69.90** | **79.27** | **56.60** | $N$ | 1 |

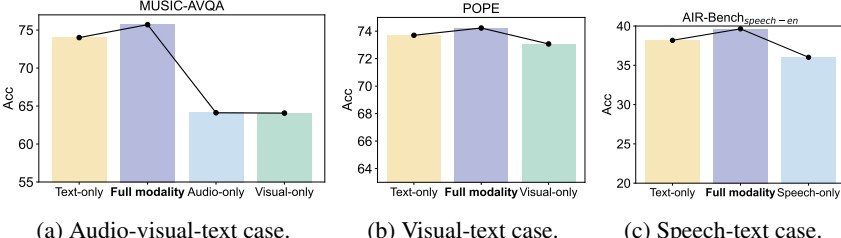

(a) Audio-visual-text case.  (b) Visual-text case.  (c) Speech-text case.

Figure 5: **Partial modality inference performance of MokA w/o cross-attention**. *Full modality*: Regular case where all multimodal tokens are processed by the MokA module w/o cross-attention. *Text-/Audio-/Visual-/Speech-only*: Only text/audio/visual/speech tokens are passed through the LoRA pathway at the first generation step during inference. Results are based on LLaMA2.

## 4.3  Partial modality inference of MokA

To further examine how effectively MokA leverages tokens from different modalities, we also conduct partial modality inference experiments where only tokens from a selected modality are passed through the LoRA adaptation pathway at the first generation during inference. It should be noted that the evaluated model is *MokA w/o cross-attention*, as cross-attention computation requires the presence of both text and non-text tokens. The results, presented in Figure 5, show that MokA w/o cross-attention significantly enhances the utilization of individual modalities compared to LoRA (as shown in Figure 1d-1f). These findings highlight that the multimodal-aware design of MokA facilitates more effective use of all available modalities.

## 4.4  Cross-modal interaction variants

In the original MokA framework, cross-attention is employed to explicitly strengthen the interaction between text and non-text tokens, thereby facilitating improved cross-modal adaptation. As previously discussed, alternative modules that similarly enhance this interaction can also be considered. In this

Table 3: Evaluation results of LoRA, its variants, and our MokA on speech-text benchmarks, $\text{MMAU}_{mini-speech}$ and $\text{AIR-Bench}_{speech-en}$. #A and #B are the number of low-rank matrices. Here $N = 2$ refers to the number of modalities.

| LLM | Method | MMAU | AIR-Bench | #A | #B |
|---|---|---|---|---|---|
| LLaMA2 | LoRA | 30.33 | 31.75 | 1 | 1 |
| | Multiple LoRA | 29.73 | 31.91 | $N$ | $N$ |
| | LoRAMoE [8] | 27.63 | 33.97 | $N$ | $N$ |
| | DoRA [21] | 26.73 | 34.36 | 1 | 1 |
| | HydraLoRA [29] | 29.13 | 31.66 | 1 | $N$ |
| | Uni-modal LoRA [1] | 38.14 | 35.14 | $N$ | $N$ |
| | Uni LoRA + MM LoRA | 37.54 | 32.56 | $N+1$ | 2 |
| | Uni LoRA + MM LoRA + Gate | 32.13 | 33.04 | $N+1$ | 2 |
| | 🍿 MokA | **38.44** | **39.64** | $N$ | 1 |
| Qwen2 | LoRA | 50.15 | 44.55 | 1 | 1 |
| | Multiple LoRA | 50.45 | 41.13 | $N$ | $N$ |
| | LoRAMoE [8] | 50.75 | 42.99 | N | N |
| | DoRA [21] | 52.55 | 44.11 | 1 | 1 |
| | HydraLoRA [29] | 53.45 | 43.94 | 1 | $N$ |
| | Uni-modal LoRA [1] | 54.05 | 47.01 | $N$ | $N$ |
| | Uni LoRA + MM LoRA | 54.16 | 43.55 | $N+1$ | 2 |
| | Uni LoRA + MM LoRA + Gate | 54.35 | 46.15 | $N+1$ | 2 |
| | 🍿 MokA | **55.26** | **49.17** | $N$ | 1 |
| LLaMA3 | LoRA | 46.25 | 43.04 | 1 | 1 |
| | Multiple LoRA | 44.74 | 43.87 | $N$ | $N$ |
| | 🍿 MokA | **51.05** | **44.39** | $N$ | 1 |

Table 4: Evaluation results of MokA and variants on audio-visual-text and visual-text cases. Results are based on LLaMA2.

| Method | Music-AVQA | AVE | $\text{MME}_{percep}$ | MMBench | POPE | SEED-Bench |
|---|---|---|---|---|---|---|
| LoRA | 73.41 | 69.84 | 908.52 | 50.64 | 70.28 | 39.71 |
| Multiple LoRA | 72.66 | 71.77 | 882.87 | 49.83 | 68.20 | 38.44 |
| Cross-attention* | 74.94 | 72.59 | 955.18 | 51.25 | 72.94 | 39.91 |
| Naive interaction | 75.04 | 73.18 | 996.73 | 51.49 | 73.52 | 40.17 |
| 🍿 MokA | 75.71 | 74.68 | 1025.86 | 52.74 | 74.23 | 40.45 |

section, we explore several variants of the cross-modal interaction module, as summarized in Table 4. The *cross-attention\** variant also adopts a cross-attention mechanism; however, it uses text tokens as queries. Consequently, the updated text tokens integrate information from the relevant non-text tokens—reversing the direction of interaction compared to the original MokA. The *naive interaction* variant performs a simple, uniform mapping from text tokens to non-text tokens without employing any attention mechanism.

Experimental results show that all proposed variants outperform the LoRA baseline, demonstrating the general effectiveness of explicitly enhancing cross-modal interactions. However, the *cross-attention\** variant performs slightly worse than the others. One possible explanation is that, unlike in other variants where text tokens remain unchanged, this variant alters text tokens by integrating non-text features. Although cross-modal interaction is enhanced, the modification of text representations may adversely affect language modeling capabilities. In addition, while *naive interaction* yields competitive results, MokA achieves further improvement through its dynamic attention mechanism. These findings suggest that the core idea of explicitly reinforcing cross-modal interactions is beneficial, and the effectiveness is not restricted to one specific module design.

### 4.5 Ablation study

To thoroughly validate the efficacy of our method, we conduct ablation studies across all three multimodal scenarios. Results are shown in Table 5. Based on the results, even without the cross-attention module, MokA w/o CA outperforms the LoRA baseline, demonstrating the effectiveness of enhancing unimodal adaptation. Furthermore, the introduction of the cross-attention module leads to additional performance improvements, indicating the benefit of explicitly enhancing cross-modal adaptation. These results indicate the necessity of each part in MokA.

Table 5: Ablation study of MokA. CA denotes Cross-Attention. Results are based on LLaMA2.

| Method | MUSIC-AVQA | POPE | AIR-Bench |
|---|---|---|---|
| LoRA [12] | 73.41 | 70.28 | 31.75 |
| Multiple LoRA | 72.66 | 68.20 | 31.97 |
| MokA w/o CA | 74.85 | 73.57 | 33.25 |
| 🍺 MokA | **75.71** | **74.23** | **39.64** |

### 4.6 Efficiency evaluation

To enable a more comprehensive comparison, we further evaluate the proposed MokA and LoRA baselines on the proportion of trainable parameters in the full model, and inference latency. As reported in Table 6, although MokA introduces additional parameters due to the inclusion of more low-rank matrices, the increase is quite modest compared to the full LLM. Also, despite

Table 6: Efficiency evaluation and performance comparison. Here, trainable parameters include low-rank matrices and all projectors. Results are based on LLaMA2.

| Method | Trainable / Total Parameters | Avg. forward time per sample | POPE Acc |
|---|---|---|---|
| LoRA [12] | 1.27% | 1.000 × | 70.28 |
| Multiple LoRA | 1.43% | 1.006 × | 68.20 |
| 🍺 MokA | 1.33% | 1.069 × | **74.23** |

MokA incurs a slight increase in inference latency compared to standard LoRA, it achieves a notable performance gain of 3.95% on the POPE benchmark. These results suggest that the additional computational cost introduced by MokA is acceptable, and the performance improvement is considerable. More detailed efficiency evaluations are provided in Appendix D.

## 5 Related works

MLLMs built upon powerful LLM backbones are increasingly demonstrating impressive capabilities across diverse downstream tasks [33, 13]. However, fine-tuning these models remains computationally expensive, prompting growing interest in parameter-efficient fine-tuning (PEFT) techniques that reduce memory and storage overhead during adaptation. Among them, LoRA has emerged as a widely adopted, and researchers have proposed several variants to further improve its efficiency and flexibility [8, 21, 29, 1]. For instance, LoRAMoE [8] introduces multiple LoRA heads combined via a gating mechanism, while DoRA [21] focuses solely on optimizing the gradient direction, enabling more efficient updates. Despite these advancements, most PEFT strategies for MLLMs are direct extensions of LLM techniques and fail to account for the inherent characteristics of multimodal learning. To address this gap, we propose MokA, a fine-tuning strategy specifically designed for MLLMs. It explicitly ensures both unimodal and cross-modal adaptation to better preserve unimodal representations and enhance cross-modal interaction, offering a targeted solution for efficient and effective multimodal adaptation.

## 6 Discussion

In this paper, we argue that both unimodal adaptation and cross-modal adaptation are essential parts for the effective fine-tuning of MLLMs, yet have largely been neglected before. To this end, we propose *Multimodal low-rank Adaptation* (**MokA**) for efficient multimodal fine-tuning. MokA redefines the roles of low-rank matrices $A$ and $B$, ensuring unimodal information is preserved while enhancing cross-modal interaction by cross-attention. We think MokA is a preliminary step toward multimodal-aware adaptation, highlighting the potential for future extensions that jointly consider both unimodal and cross-modal adaptation.

## Acknowledgement

The project was supported by the fund for building world-class universities (disciplines) of Renmin University of China, sponsored by CCF-Zhipu.AI Large Model Innovation Fund, and National Natural Science Foundation of China (NO.62106272). The project was also supported by the Shanghai Municipal Science and Technology Major Project, and Shanghai Artificial Intelligence Laboratory.

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

## A Extension to Qwen3

To further validate our MokA method, we equip it with the latest Qwen3 [35] model. Qwen3-8B is used as the LLM base model. Throughout the training process, weights of LLM are kept frozen.

Results are shown in Table 7. We conduct experiments in the audio-visual-text case. Our method can still deliver improvements compared to the LoRA baseline, with the latest Qwen3 model. These findings provide additional evidence of the effectiveness and scalability of MokA.

Table 7: Evaluation results of LoRA, its variants, and our MokA on audio-visual-text datasets, MUSIC-AVQA and AVE. #A and #B are the number of low-rank matrices. Here $N = 3$ refers to the number of modalities.

| Method | MUSIC-AVQA | AVE | #A | #B |
|---|---|---|---|---|
| LoRA | 78.57 | 74.17 | 1 | 1 |
| Multiple LoRA | 78.24 | 74.01 | $N$ | $N$ |
| 🧋 MokA | **79.54** | **75.38** | $N$ | 1 |

## B MokA with linear projection

In MokA, linear projections ($W_q$, $W_k$, and $W_v$) are not included in the cross-attention module:

$$\texttt{Att}\left(A^a\mathbf{x}^a, A^t\mathbf{x}^t, A^t\mathbf{x}^t\right) = \texttt{softmax}\left(\frac{(A^a\mathbf{x}^a)(A^t\mathbf{x}^t)^\top}{\sqrt{r}}\right)A^t\mathbf{x}^t, \tag{14}$$

$$\texttt{Att}\left(A^v\mathbf{x}^v, A^t\mathbf{x}^t, A^t\mathbf{x}^t\right) = \texttt{softmax}\left(\frac{(A^v\mathbf{x}^v)(A^t\mathbf{x}^t)^\top}{\sqrt{r}}\right)A^t\mathbf{x}^t. \tag{15}$$

The reason is that low-rank matrices $A$ of each modality actually can be considered as the linear projection in attention in this case. Therefore, we do not introduce other linear projections in the cross-attention module. In addition, projections of key and value are shared in this case ($A^t$). In fact, this kind of projection sharing strategy has been widely used to increase the efficiency of attention [15, 32]. For example, in the Linformer [32], it also utilizes the sharing key-value projection strategy to reduce computation cost. What's more, here non-text tokens are used as queries to merge textual information into non-text modalities. In this way, cross-modal interaction is explicitly enhanced, while text tokens are kept unchanged to avoid potential disruption to the model's original strong text understanding capability.

To further validate the idea of cross-attention, we conduct experiments that include linear projections in cross-attention. The concrete attention mechanism with linear projection is conducted as follows, and the notation is consistent with the main manuscript:

$$\texttt{Att}\left(A^a\mathbf{x}^a, A^t\mathbf{x}^t, A^t\mathbf{x}^t\right) = \texttt{softmax}\left(\frac{(W_q^a A^a\mathbf{x}^a)(W_k^t A^t\mathbf{x}^t)^\top}{\sqrt{r}}\right)W_v^t A^t\mathbf{x}^t, \tag{16}$$

$$\texttt{Att}\left(A^v\mathbf{x}^v, A^t\mathbf{x}^t, A^t\mathbf{x}^t\right) = \texttt{softmax}\left(\frac{(W_q^v A^v\mathbf{x}^v)(W_k^t A^t\mathbf{x}^t)^\top}{\sqrt{r}}\right)W_v^t A^t\mathbf{x}^t. \tag{17}$$

Here, $W_q^a$ is the linear projection of the audio query, and the others are similar.

In Table 11, we provide the results in the audio-visual-text and visual-text cases. Based on the results, MoKA with linear projection, yields improvements compared with LoRA baseline. However, it does not consistently outperform the original MoKA and introduces additional trainable parameters along with increased computational overhead.

## C More than three modality case

In this section, we extend our experiments to scenarios with more than three modalities. Specifically, we consider a four-modality setting involving audio, visual, point cloud, and language data, and

evaluated it on the MCUB-3 benchmark [3]. In this 3+ modality case, LoRA has an accuracy of 37.41%. Our MokA has an accuracy of 45.58%, indicating its scalability and effectiveness under 3+ modality cases.

# D   Efficiency evaluation

Compared to LoRA, MokA introduces additional A matrices and a cross-attention module. However, it should be noted that the additional computational cost of MokA only comes from the cross-attention module, which is flexible and can be replaced by a more efficient strategy if needed. In this section, we provide a more detailed efficiency analysis comparing FLOPs, GPU memory usage, and average forward time per sample (proportional to training time). For clarity, we report these metrics for the two-modality (VL), three-modality (AVL), and four-modality (AVPL) settings. As shown in the tables, MokA's extra computational and memory cost remains limited and acceptable for typical multimodal scenarios.

Table 8: Efficiency evaluation between LoRA and MokA on various datasets. Results are based on LLaMA2.

| VL (MME$_{percep}$) | FLOPs | Memory Usage | Avg. forward time/sample |
|---|---|---|---|
| LoRA | 1.000x | 1.000x | 1.000x |
| 🍿 MokA | 1.009x | 1.001x | 1.069x |

| AVL (MUSIC-AVQA) | FLOPs | Memory Usage | Avg. forward time/sample |
|---|---|---|---|
| LoRA | 1.000x | 1.000x | 1.000x |
| 🍿 MokA | 1.021x | 1.001x | 1.134x |

| AVPL (MCUB-3) | FLOPs | Memory Usage | Avg. forward time/sample |
|---|---|---|---|
| LoRA | 1.000x | 1.000x | 1.000x |
| 🍿 MokA | 1.013x | 1.002x | 1.213x |

# E   Audio-visual interaction

In the original MokA, only the attention between text and non-text tokens is considered. It is motivated by the fact that text modality typically conveys the question or task description, while the audio and visual modalities provide environmental information, in the instruction. Therefore, cross-attention is applied to explicitly enhance the interaction between the task (text token) and environment (non-text token). It is also worth exploring whether further interactions between the scene modalities themselves—i.e., audio and visual—can be beneficial. To this end, we conduct additional ablation studies. Experiments are conducted based on LLaMA2. The table reports the results for audio-visual attention with both audio as query and video as query. The results show that introducing additional audio-visual attention can bring gains, but it is not very noticeable. The potential benefit from further enhancing explicit audio-visual interactions is relatively limited.

Table 9: Evaluation results of LoRA and MokA variants on audio-visual-text datasets. Results are based on LLaMA2.

| Method | MUSIC-AVQA | AVE |
|---|---|---|
| LoRA | 73.41 | 69.84 |
| 🍿 MokA | 75.71 | 74.68 |
| MokA w/ audio query att. | 75.78 | 74.53 |
| MokA w/video query att. | 75.76 | 74.81 |

# F Ablation study of rank

In this section, we conduct experiments of MokA with different ranks. It consistently outperforms the LoRA baseline across different rank settings.

Table 10: Evaluation results of LoRA and MokA on MUSIC-AVQA and AVE with different ranks. Results are based on LLaMA2.

| Method | MUSIC-AVQA | AVE | Rank |
|--------|-----------|-----|------|
| LoRA | 73.41 | 69.84 | r=4 |
| LoRA | 73.56 | 70.01 | r=8 |
| LoRA | 73.73 | 70.07 | r=12 |
| 🍿 MokA | 75.71 | 74.68 | r=4 |
| 🍿 MokA | 74.68 | 74.71 | r=8 |
| 🍿 MokA | 74.89 | 74.36 | r=12 |

# G Case study of cross-attention in MokA

In this section, we conduct a case study on the cross-attention module in MokA. This module explicitly integrates task description information from text tokens with non-text tokens, thereby facilitating cross-modal interaction. Here we provide two samples from an audio-visual-text scenario, as shown in Figure 6. The visualization shows the cross-attention weights of the $q_{proj}$ at the $10-$th layer. The LLM model is LLaMA2. The results indicate that during the process of explicit cross-modal integration (e.g., cross-attention), text tokens that are more related to a given modality can receive higher attention weights. For example, the token "sound" receives greater attention in relation to the audio modality. This cross-modal integration can better facilitate the alignment and interaction between text tokens and non-text tokens.

Table 11: Experiments of MokA with linear projection under the audio-visual-text and visual-text cases. The LLM backbone is LLaMA2.

| Method | Music-AVQA | $MME_{percep}$ | MMBench | POPE | SEED-Bench |
|--------|-----------|----------------|---------|------|------------|
| LoRA | 73.41 | 908.52 | 50.64 | 70.28 | 39.71 |
| Multiple LoRA | 72.66 | 882.87 | 49.83 | 68.20 | 38.44 |
| MokA w/ linear projection | 73.83 | 926.77 | 53.97 | 72.43 | 41.01 |
| 🍿 MokA | 75.71 | 1025.86 | 52.74 | 74.23 | 40.45 |

# H Broader impacts

In this paper, we aim to contribute to the efficient fine-tuning of MLLM, particularly how they well process and integrate information from different modalities. Improvements in this area may support downstream applications in fields like autonomous driving and education. At the same time, this line of research carries certain risks. For example, there is a possibility that MLLM could reflect or amplify biases present in the training data, or be misused in sensitive contexts. We do not directly address these issues in this work, but acknowledge them as important areas for future research. All datasets used are publicly available, and we follow standard filtering procedures to reduce exposure to harmful content.

# I Datasets

Information about the datasets used in our experiments is provided in this section.

Video-LLaVA [19] used a mixed dataset of images and videos for video captioning and image captioning tasks. The dataset includes a 665k image-text instruction and a 100k video-text instruction. This dataset is used for pre-training the visual branch.

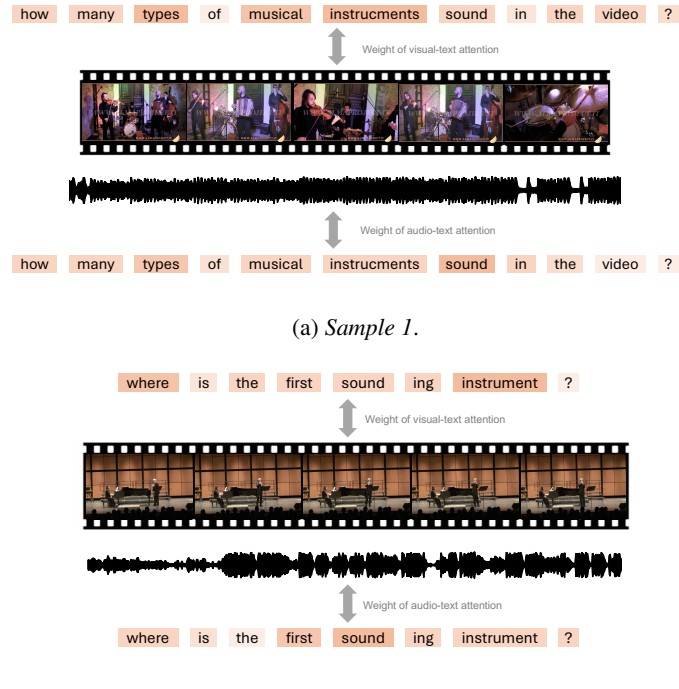

(a) *Sample 1.*

(b) *Sample 2.*

Figure 6: Cross-attention weight visualization, where deeper colors indicate higher attention weights.

AudioCaps [14] dataset is used for the audio captioning task. It consists of 46K pairs of audio clips and text descriptions. This dataset is used for pre-training the audio branch.

GigaSpeech-M [4] is a 1000h dataset for speech recognition task. This dataset is used for pre-training the speech branch.

MUSIC-AVQA [17] is an audio-visual-text dataset, which is introduced to support spatio-temporal understanding of musical content. It offers 45K QA pairs across 33 question templates that span multiple modalities and question types.

AVE [30] is an audio-visual-text dataset. It focuses on the audio-visual event localization task. This dataset covers 28 event classes and consists of 4,143 samples.

LLaVA-Instruct-150K [20] is a set of GPT-generated multimodal instruction-following data. It is used for instruction fine-tuning for the visual-text case.

LibriSpeech [23] is a 960-hour dataset. We use the instruction from [28] for instruction fine-tuning of the speech-text case.

$MME_{percep}$ [9] is the perception subset of the MME benchmark, covering a total of 10 subtasks for the evaluation of the visual-text perception ability.

MMBench [22]is a collection of benchmarks to evaluate the visual-text understanding capability. It has 3,000 multiple-choice questions covering object detection, text recognition, action recognition, image captioning, relation reasoning, and so on.

POPE [18] is a benchmark that is used for evaluating the visual-text understanding ability of MLLM. The used image is the test set of MSCOCO dataset.

SEED-Bench [16] consists of 19K multiple-choice questions with accurate human annotations for evaluating the visual-text understanding ability of MLLM.

$MMAU_{mini-speech}$ [27] is the speech subset of MMAU-mini benchmark. This benchmark is used for evaluating the speech-text understanding ability of MLLM.

$AIR-Bench_{speech-en}$ [37] is the English speech subset of the foundation part of AIR-Bench. This benchmark is used for evaluating the speech-text understanding ability of MLLM.

