# OpenReview forum: "MokA: Multimodal Low-Rank Adaptation for MLLMs"
_NeurIPS.cc/2025/Conference — NeurIPS 2025 oral_

### Official Review · Reviewer_3Vnj · 2025-06-25

**Clarity:** 4
**Significance:** 4
**Originality:** 3
**Rating:** 6
**Confidence:** 4

**Summary:**

This paper introduces a new mechanism for multimodal low-rank adaptation for MLLMs. In a nutshell, conventional LoRA uses matrix A to project down to a low-rank space, then uses matrix B to project to original dimension. And for multimodal LLMs, A and B are shared across 3 modalities: text, audio, and vision.

This paper proposes to: (1) use 3 different A matrices, one for each modality. (2) use a cross-attention among the 3 modalities in the projected space, and update the embeddings for audio and vision. (3) use a shared B matrix.

**Questions:**

Line 219: "a baseline that uses 3 A matrices and B A matrices" - what does this mean? If it's a typo, please fix.

Table 4: Why is "Multiple LoRA" worse? And why is "MokA w/o CA" better? What are the major differences? Is sharing or not sharing B matrix the only difference?

Section 2.2.2: So this cross attention mechanism does not introduce any new parameters, and only introduce new computes? If my understanding is correct, this is really worth highlighting (both for the audience to understand, and for emphasizing as a strength of the method).

Section 2.2.2: So Query is from audio/vision, Key is from text, and Value is from text? Does this use any positional encoding?

Which part of the LLM do we apply LoRA or MokA in the experiments? In another word, what is the "W" matrix that you are adapting in the experiments? Did you adapt all the weight matrices?

**Ethical Concerns:**

["NO or VERY MINOR ethics concerns only"]

**Final Justification:**

The authors have addressed all of my raised concerns. We stick with my original rating.

**Limitations:**

The cross attention vs causal decoding seems an important limitation here, especially for MLLMs generating multimodal tokens. The paper did not cover or discuss them.

**Paper Formatting Concerns:**

No formatting concerns.

**Quality:**

4

**Strengths And Weaknesses:**

Strengths:

Paper is well written. Very easy to follow. Very readable.

Figure 1 is a great example showcasing the problem of conventional LoRA on multimodal tasks: "LoRA parameters is overly influenced by text tokens". I am very convinced by how this was presented.

The math part is also very clear. All variables are clearly defined. Equations are easy to follow and all technically correct.

Table 1 did a great job explaining the differences between alternative approaches. Using #A and #B makes it very easy to understand.

The proposed method outperforms alternative methods almost on all multimodal setups.

Weaknesses:

The efficiency evaluation in Section 4.5 looks a bit problematic to me, especially the Inference Latency part. How is latency measured? What is the unit? Why Multiple LoRA has more latency than LoRA? To me it seems they should be same, if the implementations are not too different. MokA should have the highest latency because the computation of the cross attention is expensive. If the implementations are not apple-to-apple comparable, I would suggest using FLOPs instead of latency.

The paper did not cover an important technical detail: how does decoding work with MokA, especially with the cross attention? By introducing cross attention, the model is no longer causal. Every time when the model decodes a new text token, do we recompute the projected embeddings for all audio/vision tokens? Or the cross attention only applies to the prompt part, not the decoded / response part? What about MLLMs that are capable of generating audio/vision tokens?

---

> ### Author Rebuttal · Authors · 2025-07-31
>
> We sincerely thank the reviewer for the positive feedback on our work. We also appreciate the helpful suggestions regarding the efficiency evaluation, the discussion of the cross-attention module, and other valuable advice. In response, we will include related experiments and discussions in the revised manuscript to make the paper clearer.
>
> ### **1. Efficiency evaluation**
>
> In the previous implementation of Multiple-LoRA, we did not merge the multiple LoRA heads during inference, to keep the implementation modular easily extensible to other LoRA variants. As a result, each token passes through multiple A and B matrices independently, bringing additional cost. This could be the reason that Multiple-LoRA has a longer latency in the previous experiments.
>
> Thank you for pointing this out. We improve the implementation of Multiple-LoRA, merging multiple LoRA heads during inference. In addition, we also add additional efficiency metrics, including FLOPs and GPU memory usage, to better evaluate the efficiency. The following table shows results under Vision-Language case as an example.
>
> Based on the results, Multiple-LoRA-merged during inference has a similar computational cost. MokA obtains noticeable performance improvement with only modest extra resource consumption. We will add these experiments and discussion to the revised manuscript.
>
> ---
>
> | **VL (MME_${percep}$)** | **FLOPs** | **Memory Usage** | **Avg. forward time/sample** |
> | --- | --- | --- | --- |
> | LoRA | 1526.25 GFLOPs (1.000x) | 27315.75 MB  (1.000x) | 1.75 s (1.000x) |
> | Multiple LoRA-merged | 1526.67 GFLOPs (1.000x) | 27349.53 MB (1.001x) | 1.76 s (1.006x) |
> | MokA | 1539.55 GFLOPs (1.009x) | 27329.78 MB (1.001x) | 1.87 s (1.069x) |
>
> ---
>
> ### **2. Discussion about cross-attention module**
>
> Here we will provide a more detailed discussion about our cross-attention module.
>
> 1. **Cross-attention only applies to prompt tokens**
>
>     In the original MokA, cross-attention is motivated by the fact that text modality typically conveys the question or task description, while the audio and visual modalities provide environment information, in the prompt. Cross-attention is applied to explicitly enhance the interaction between the **task (text token in the prompt) and environment (non-text token in the prompt).**
>
>     **Therefore, the cross-attention is only applied to the prompt parts.** Concretely, during training, only the prompt tokens (excluding response tokens) are considered in the cross-attention module. During inference, cross-attention only operates on the prefilling stage, where the model processes the entire prompt sequence and generates the first output token. The decoding stage is not applied.
>
>     Accordingly, **MokA does not greatly break the causal nature when generating response. It only enhances the comprehension of prompt information.**
>
> 2. **Query and positional encoding**
>
>     Yes, Query is from audio/vision tokens. Key and Value are from text tokens. Regarding positional encoding, we retain the original positional encoding of the LLMs, and do not introduce any new encoding specifically for the cross-attention for efficiency.
>
> 3. **No additional parameters**
>
>     Yes, this module does not introduce any new parameters, and only introduces new computes. Before passing into the cross-attention module, tokens of each modality are processed by their own low-rank matrices $A$. **The low-rank matrices $A$ of each modality can be approximately regarded as the linear projection in the standard attention in this case.** In addition, projections of key and value are shared in this case (since key and value are both text tokens that are processed by $A^t$). And this kind of projection sharing strategy has been shown to be effective for improving attention efficiency [1]. **Therefore, we do not introduce other linear projections in the cross-attention module to be more efficient.** Thank you for this helpful advice, and we will add the related discussion to the revised manuscript.
>
>
> [1] Sinong Wang, Belinda Z Li, Madian Khabsa, Han Fang, and Hao Ma. Linformer: Self-attention with linear complexity. arXiv preprint arXiv:2006.04768, 2020.
>
> ---
>
> ### **3. MLLMs can generate audio/vision tokens**
>
> At present, our work focuses on scenarios where the MLLMs generate text tokens while conditioning on audio/vision/text inputs. MLLMs that can generate audio/vision tokens are indeed an important and exciting direction for future research. We sincerely thank the reviewer for inspiring us to consider this promising direction.
>
> ---
>
> ### **4. Discussion about Multiple-LoRA & MokA**
>
> Here we first provide the differences between LoRA, Multiple-LoRA, and MokA w/o CA. Suppose the number of modalities is $N$:
>
> - **LoRA**: uses one matrix A and one matrix B, both shared by all modalities.
> - **Multiple-LoRA**: uses $N$ matrices A and $N$ matrices B. But all matrices (both A and B) are still fully shared across modalities.
> - **MokA w/o CA**: uses $N$ modality-specific matrices A but one shared matrix B. Here, the matrices A adapt each modality independently, while the shared matrix B maps the separated unimodal subspaces back into a common latent space.
>
> **Multiple-LoRA** simply expands standard LoRA’s capacity by adding multiple heads but still treats all modalities the same—ignoring their unique characteristics. However, just increasing capacity while fully sharing parameters across all modalities does not solve the specific challenges of multimodal scenarios. It still does not consider both unimodal adaptation and multimodal adaptation.
>
> In contrast, **design of MokA w/o CA is multimodal-aware. It explicitly ensures better unimodal adaptation. MokA w/o CA explicitly captures unimodal information by introducing modality-specific matrices A**. It ensures that each modality’s unique information is individually captured before merging. This simple but crucial design helps better utilize unimodal adaptation. This explains why MokA w/o CA outperforms Multiple-LoRA, even with fewer total parameters.
>
> ---
>
> ### **5. Applied weights of LoRA/MokA**
>
> In experiments, we apply LoRA or MokA modules to important weights within each transformer block. Specifically, we insert them into:
>
> - q_proj, k_proj, v_proj, o_proj (the self-attention projection weights),
> - gate_proj, up_proj, and down_proj (the MLP feed-forward layers).
>
> ---
>
> ### **6. Typos**
>
> We thank the reviewer for pointing out the typos. We have carefully revised the manuscript to correct these errors. Thank you for helping us improve the clarity and presentation of the paper.

---

> > ### Comment · Reviewer_3Vnj · 2025-08-04
> >
> > The authors' responses are satisfactory.
> >
> > I would suggest the authors to include relevant information in the revised manuscript, especially (1) the updated table on flops/latency, (2) highlighting cross-attention only applies to prompt, (which is also a potential limitation for multimodal output) (3) clarification on which weights are adapted.

---

> > > ### Author Response · Authors · 2025-08-04
> > >
> > > Thanks for the constructive comments, which definitely help us polish this paper! We will include relevant information in the revised manuscript as suggested.

---

> ### Author Response · Authors · 2025-08-04
>
> Dear reviewer, we would like to know if our responses have successfully resolved your concerns. We are willing to respond to any further questions.

---

### Official Review · Reviewer_DUve · 2025-06-29

**Clarity:** 4
**Significance:** 3
**Originality:** 3
**Rating:** 5
**Confidence:** 4

**Summary:**

This paper proposes Multimodal low-rank Adaptation (MokA), consisting of unimodal matrix A, task-centric cross-attention, and shared multimodal matrix B. MokA provides a more targeted solution for efficient adaptation of MLLMs instead of borrowing from LLM. Experiments demonstrate that MokA achieves better performance across multiple benchmarks.

**Questions:**

1. Are all the methods listed in the table trained under the same settings?
2. Since tokens from different modalities are processed by separate matrices A, does MokA mitigate catastrophic forgetting of the LLM’s pretrained knowledge? To verify, for example, this is particularly relevant for the MME reasoning subset, which relies heavily on the model’s original language capabilities. Could you report the performance on the MME reasoning subset specifically?

**Ethical Concerns:**

["NO or VERY MINOR ethics concerns only"]

**Final Justification:**

The author supplements hyperparameter experiments and related work comparisons in the rebuttal, which address all of my concerns. Given the clear motivation and technical contribution, I would recommend accepting this paper. However, since this paper does not bring breakthrough progress or significant performance improvement, I am unable to continue increasing my score.

**Limitations:**

yes

**Quality:**

3

**Strengths And Weaknesses:**

### Strengths

1. **Clear motivation.** This paper is motivated by designing an efficient fine-tuning strategy tailored for the multimodal nature of MLLMs. A preliminary study clearly demonstrates the limitations of the existing unimodal PEFT strategies.
2. **Technical contributions.** This paper innovatively decouples the projection matrices A and B and designs a cross-modal interaction mechanism between them.
3. **Comprehensive experiments.** This paper conducts experiments across audio-visual-text, visual-text, and speech-text benchmarks, and makes comparisons with a variety of PEFT methods.

### Weaknesses

1. **Lack of discussion regarding hyperparameters.** The rank of low-rank matrices is 4, which is not consistent with the optimal rank 8 reported in the LoRA paper. How was this parameter selected, and why not experiment with the optimal configuration?
2. **Limited related works discussion.** There are several existing works, like Omni-SMoLA[1] already propose PEFT methods suitable for multimodal learning by assigning tokens from different modalities to different LoRAs. However, this paper only discusses methods that are not designed for multimodal learning.

---

[1] Wu, JIalin, et. al. “Omni-SMoLA: Boosting Generalist Multimodal Models with Soft Mixture of Low-rank Experts”

---

> ### Author Rebuttal · Authors · 2025-07-31
>
> We are deeply grateful to the reviewer’s positive feedback and helpful comments. Based on suggestions, we have added new ablation studies, comparisons, and evaluations, which will be included in the revised manuscript.
>
> ### **1. Ablation studies about rank**
>
> Thank you for the valuable suggestion. In our experiments, we did not specifically tune the rank hyperparameter, as our primary focus is to provide a multimodal-aware PEFT method that is rank-agnostic. Based on this suggestion, we conducted ablation studies with different ranks, and the results are shown in the following table. Experiments are conducted based on LLaMA2. These results indicate that simply increasing the rank does not always lead to performance gains, which aligns with the observation in the original LoRA paper [1]: “*We argue that increasing r does not cover a more meaningful subspace, which suggests that a low-rank adaptation matrix is sufficient.*”
>
> More importantly, MokA consistently outperforms the LoRA baseline across different rank settings, demonstrating its scalability. We will include this discussion and the new results in the revised manuscript. Thank you again for highlighting this point.
>
> ---
>
> |  |  | **MUSIC-AVQA** | **AVE** | **MME_${percep}$** | **MMBench** | **POPE** | **SEED-Bench** |
> | --- | --- | --- | --- | --- | --- | --- | --- |
> | LoRA | r=4 | 73.41 | 69.84 | 908.52 | 50.64 | 70.28 | 39.71 |
> | LoRA | r=8 | 73.56 | 70.01 | 947.24 | 51.23 | 70.63 | 39.57 |
> | LoRA | r=12 | 73.73 | 70.07 | 916.76 | 51.37 | 70.17 | 39.46 |
> | MokA | r=4 | 75.71 | 74.68 | 1025.86 | 52.74 | 74.23 | 40.45 |
> | MokA | r=8 | 74.68 | 74.71 | 1027.84 | 52.83 | 74.43 | 40.38 |
> | MokA | r=12 | 74.89 | 74.36 | 1007.33 | 52.81 | 74.18 | 40.97 |
>
> ---
>
> [1] LoRA: Low-Rank Adaptation of Large Language Models, ICLR, 2022.
>
> ---
>
> ### **2. Comparison to related methods**
>
> We thank the reviewer for pointing out the related method, Omni-SMoLA.
>
> **Compared to MokA, Omni-SMoLA adopts a more complex design with noticeably more parameters**. For example, in an AVL (Audio-Visual-Language) scenario, Omni-SMoLA employs separate SMoLA modules for each modality (visual, audio, and text), in addition to an extra multimodal SMoLA. Each SMoLA module internally integrates multiple LoRA branches.
>
> In contrast, MokA only uses a few low-rank A matrices (one per modality) and a single shared low-rank B matrix, without stacking multiple independent LoRA blocks. Therefore, Omni-SMoLA can have several times more parameters than MokA.
>
> Concretely, in experiments, we set each SMoLA to have two LoRA branches. Therefore, in the AVL case, Omni-SMoLA has four SMoLA modules (audio, visual, text and multimodal), and each SMoLA has two LoRA branches. Then the total number of low-rank matrices (both A and B) is 16, which is much more than MokA. The VL case is similar.
>
> The comparison results in the following table show that **MokA achieves comparable or even better performance than Omni-SMoLA, while maintaining a simpler and more lightweight design**. Experiments are conducted based on LLaMA2.
>
> We will add this comparison and discussion to the revised manuscript.
>
> ---
>
> | **AVL case** | **MUSIC-AVQA** | **AVE** | **#A** | **#B** |
> | --- | --- | --- | --- | --- |
> | LoRA | 73.41 | 69.84 | 1 | 1 |
> | MokA | 75.71 | 74.68 | 3 | 1 |
> | Omni-SMoLA | **75.73** | **74.70** | 8 | 8 |
>
> ---
>
> | **VL case** | **MME_${percep}$** | **MMBench** | **POPE** | **SEED-Bench** | **#A** | **#B** |
> | --- | --- | --- | --- | --- | --- | --- |
> | LoRA | 908.52 | 50.64 | 70.28 | 39.71 | 1 | 1 |
> | MokA | **1025.86** | 52.74 | **74.23** | 40.45 | 2 | 1 |
> | Omni-SMoLA | 972.64 | **53.01** | 73.43 | **41.68** | 6 | 6 |
>
> ---
>
> ### **3. Evaluation on MME reasoning benchmark**
>
> We thank the reviewer for this helpful suggestion to evaluate performance on the MME reasoning benchmark to assess potential catastrophic forgetting of the LLM’s pretrained knowledge. Following this advice, we conduct an evaluation on the MME reasoning benchmark. Experiments are conducted based on LLaMA2.
>
> On the MME reasoning benchmark, the LoRA baseline achieves an accuracy of 16.78%, while MokA reaches 18.01%, indicating that MokA has stronger capacity in this aspect.
>
> It should be noted that both LoRA and MokA do not obtain high scores on this benchmark, as our work focuses only on the supervised fine-tuning (SFT) stage and does not include additional designs specifically aimed at enhancing reasoning ability. Nevertheless, under the same setting, MokA achieves consistently better results, demonstrating its effectiveness.
>
> We will include the new MME reasoning results and discussion in the revised manuscript.
>
> ---
>
> ### **4. Training setting**
>
> Yes, all the compared methods under the same scenarios are conducted under the same settings to ensure fairness.

---

> > ### Comment · Reviewer_DUve · 2025-08-06
> >
> > Thank you for the detailed response. The rebuttal addressed my concerns, so I will be raising my score.

---

> > > ### Author Response · Authors · 2025-08-06
> > >
> > > Thank you for your valuable and helpful suggestions! We will incorporate these experiments and discussions into the revised version of the manuscript to enhance its quality.

---

> ### Author Response · Authors · 2025-08-04
>
> Dear reviewer, we would like to know whether our responses addressed your concern. We are happy to answer any further questions and comments.

---

### Official Review · Reviewer_x9zT · 2025-07-02

**Clarity:** 3
**Significance:** 2
**Originality:** 2
**Rating:** 4
**Confidence:** 3

**Summary:**

This paper advances along the LoRA direction, proposing a targeted variant, MokA, for Multimodal-LLM. The variant is motivated by the observation that the update of vanilla LoRA is dominated by text tokens. Accordingly, the author allocates modality-specific LoRA A parameters to decouple their update. Additionally, cross-modal communication is injected within the MokA module.

**Questions:**

Questions:

1. Considering the LoRA baseline, your ablations have already shown that 'Multiple LoRA' does not enhance performance, but I wonder if increasing the rank of LoRA baseline could lead to better results? I mean, if in practice I find my LoRA model not performing well, my first reaction would be to try increasing the hidden rank instead of trying a much more complicated variant.
2. "Specifically, we evaluate the model’s performance when only tokens from a selected modality are passed through the LoRA adaptation pathway at the first generation step during inference. "  Could the author clarify what the highlighted "at the first generation step during inference" means?  I guess this means something similar to "during prompt prefilling" or "during building the KV cache", but I am not sure.
3. The comparison result between "Multiple-LoRA" and "MokA w/o CA" is counterintuitive to me. The result means that a shared B matrix outperforms multiple modality-specific B matrices. Could the author provide some analysis/explanation?
3. Arrows in Fig.4 are messy, and the lines cross arbitrarily. I strongly suggest refining this figure
5. While the "Uni LoRA + MM LoRA" and "Uni LoRA + MM LoRA + Gate" baselines are interesting, their relationship with the proposed MokA is not sufficiently explained in the paper. What do the authors intend to verify by introducing these baselines?

Overall, I would be happy to raise my score if some of the above questions can be satisfactorily answered.

**Ethical Concerns:**

["NO or VERY MINOR ethics concerns only"]

**Final Justification:**

After the discussion, this paper is complete and sound to me. I am therefore happy to raise my score.

Meanwhile, since I have not been closely tracing the recent literature on this topic, for novelty and comparative experiments, I would suggest the AC to give more weight to the opinions of other reviews.

**Limitations:**

See questions

**Quality:**

3

**Strengths And Weaknesses:**

Strengths:

- The paper has a clear logical flow, which makes it easy to follow.
- Experiment designs are reasonable and well support the paper's claims

Weaknesses:
- The proposed method breaks the simplicity of LoRA: the side branch can no longer be merged into the main LLM parameters, and the introduced cross-attention module adds additional complexity conceptually, engineerly, and computationally.

---

> ### Author Rebuttal · Authors · 2025-07-31
>
> We greatly appreciate the reviewer for the constructive and detailed comments, which help us improve the quality, clarity and presentation of the manuscript. We will add these experiments and analysis to the revised manuscript.
>
> ### **1. Ablation studies about rank**
>
> Based on the suggestions, we explore the effect of varying the rank. Experiments are conducted based on LLaMA2. As shown in the following table, **simply increasing the rank does not necessarily lead to consistent performance gains.** This observation aligns with the finding reported in the original LoRA paper [1]: “*We argue that increasing r does not cover a more meaningful subspace, which suggests that a low-rank adaptation matrix is sufficient.*”
>
> More importantly, our observation in Figure 1 of the manuscript shows that the optimization of all-modality-shared LoRA parameters is overly influenced by text tokens, resulting in non-text tokens being less effectively utilized. Therefore, **even with an increased rank, the parameters remain fully shared across modalities, which still structurally prevents independent utilization of unimodal information.** The issue is not just about capacity, but more about whether the module structure can effectively capture modality-specific characteristics and cross-modal interactions.
>
> Based on this, **MokA is designed to explicitly introduce modality-specific parameters and cross-modal interaction, addressing this structural limitation that increasing rank alone cannot resolve.** We also conduct experiments of MokA with different ranks. It consistently outperforms the LoRA baseline across different rank settings.
>
> ---
>
> |  |  | **MUSIC-AVQA** | **AVE** | **MME_${percep}$** | **MMBench** | **POPE** | **SEED-Bench** |
> | --- | --- | --- | --- | --- | --- | --- | --- |
> | LoRA | r=4 | 73.41 | 69.84 | 908.52 | 50.64 | 70.28 | 39.71 |
> | LoRA | r=8 | 73.56 | 70.01 | 947.24 | 51.23 | 70.63 | 39.57 |
> | LoRA | r=12 | 73.73 | 70.07 | 916.76 | 51.37 | 70.17 | 39.46 |
> | MokA | r=4 | 75.71 | 74.68 | 1025.86 | 52.74 | 74.23 | 40.45 |
> | MokA | r=8 | 74.68 | 74.71 | 1027.84 | 52.83 | 74.43 | 40.38 |
> | MokA | r=12 | 74.89 | 74.36 | 1007.33 | 52.81 | 74.18 | 40.97 |
>
> ---
>
> [1] LoRA: Low-Rank Adaptation of Large Language Models, ICLR, 2022.
>
> ---
>
> ### **2. First token generation process**
>
> Here, “at the first generation step during inference” refers specifically to the prefilling stage, where the model processes the entire prompt sequence and generates the first output token.
>
> Thank you for pointing this out, and we will revise the related parts of the manuscript to make it more readable.
>
> ---
>
> ### **3. Discussion about multiple-LoRA & MokA w/o CA**
>
> To clarify, the following explains the differences between LoRA, Multiple-LoRA, and MokA w/o CA. Suppose the number of modalities is $N$:
>
> - **LoRA**: uses one matrix A and one matrix B, both shared by all modalities.
> - **Multiple-LoRA**: uses $N$ matrices A and $N$ matrices B. But all matrices (both A and B) are still fully shared across modalities. *There are no multiple modality-specific matrices.* The multiple heads simply increase capacity without distinguishing modality-specific characteristics.
> - **MokA w/o CA**: uses $N$ modality-specific matrices A and one shared matrix B. Here, the matrices A adapt each modality independently, while the shared matrix B maps the separated unimodal subspaces back into a common latent space.
>
> **Multiple-LoRA** can be viewed as simply expanding the capacity of standard LoRA by adding multiple heads. However, *it still treats all modalities identically, i.e. all parameters are shared across all modalities*. There are no modality-specific matrices in this method. In this fully shared case, as discussed above, optimization of parameters is overly influenced by text tokens, which results in non-text tokens being less effectively utilized. Therefore, simply increasing the number of LoRA heads does not provide a significant benefit in addressing the multimodal challenge.
>
> By contrast, **MokA w/o CA** explicitly captures unimodal information by introducing modality-specific matrices *A*. This design better captures unique modality characteristics before merging them back with the shared B matrix.
>
> Therefore, the better performance of MokA w/o CA comes from its more effective use of modality-specific information.
>
> ---
>
> ### **4. Discussion about "Uni LoRA + MM LoRA" and "Uni LoRA + MM LoRA + Gate"**
>
> As mentioned in Sec 2.1, one of the core motivations for our work is that existing LoRA variants for multimodal adaptation often overlook the need to jointly address both unimodal adaptation and cross-modal adaptation. Our MokA method is proposed to address this limitation.
>
> Since most LoRA variants do not explicitly consider multimodal characteristics, **we further introduce two more relevant and competitive baselines to provide a more comprehensive comparison.**
>
> Concretely, these two baselines, Uni LoRA + M LoRA and Uni LoRA + MM LoRA + Gate, are straightforward designs that aim to address both unimodal and multimodal adaptation:
>
> - “Uni LoRA + MM LoRA” directly uses unimodal LoRA to fit unimodal adaptation, and multimodal LoRA to fit multimodal adaptation.
> - “Uni LoRA + MM LoRA + Gate” further adds a gating mechanism to adaptively integrate the unimodal and multimodal branches.
>
> **These baselines demonstrate that introducing a multimodal-aware design—i.e., accounting for both unimodal and cross-modal adaptation—can indeed lead to improved performance, confirming that jointly considering both aspects is necessary.**
>
> However, these variants rely on simply stacking separate LoRA modules and thus require substantially more parameters.
>
> In contrast, **MokA** achieves both unimodal and multimodal adaptation more neatly and efficiently through its modality-specific A matrices and flexible cross-attention module, **achieving better performance with a simpler and more cohesive structure**.
>
> ---
>
> ### **5. Discussion about the simplicity of LoRA**
>
> Thank you for pointing this out. Indeed, MokA makes the side branch separate from the main LLM parameters and thus it can no longer be merged back. However, **multimodal scenarios are inherently more complex, and it is often inevitable to introduce additional cost to better capture multimodal characteristics.**
>
> We have aimed to keep the structure of MokA as neat and efficient as possible. Based on our efficiency evaluation in terms of FLOPs, GPU memory usage, and average forward time per sample, MokA introduces only modest additional resource consumption while achieving significant performance gains. Therefore, we believe MokA is practical with noticeably better performance.
>
> ---
>
> | **VL (MME_${percep}$)** | **FLOPs** | **Memory Usage** | **Avg. forward time/sample** |
> | --- | --- | --- | --- |
> | LoRA | 1526.25 GFLOPs (1.000x) | 27315.75 MB  (1.000x) | 1.75 s (1.000x) |
> | MokA | 1539.55 GFLOPs (1.009x) | 27329.78 MB (1.001x) | 1.87 s (1.069x) |
>
> ---
>
> | **AVL (MUSIC-AVQA)** | **FLOPs** | **Memory Usage** | **Avg. forward time/sample** |
> | --- | --- | --- | --- |
> | LoRA | 12116.72 GFLOPs  (1.000x) | 27961.74 MB (1.000x) | 5.53 s (1.000x) |
> | MokA | 12375.90 GFLOPs (1.021x) | 27996.49 MB (1.001x) | 6.27 s  (1.134x) |
>
> ---
>
> ### **6. Enhancing Fig. 4**
>
> Thank you for this suggestion. We will revise Fig.4 to make the arrows/lines clearer and more organized, improving its clarity.

---

> > ### Comment · Reviewer_x9zT · 2025-08-03
> >
> > Thanks for the clarification. I am mostly satisfied with the rebuttal.
> >
> > Concerning "multiple lora," I suggest refining the manuscript on this point, since readers will very likely interpret this as the "modality-specific lora" setting.
> >
> > I have a follow-up question:
> >
> > MLLMs typically employ a uni-directional attention mask. This means that for an input like your example:
> >
> > ```
> > <audio> <visual> Please answer the question: which clarinet makes the sound first?
> > ```
> > The audio and visual tokens would originally be unable to receive any information from the following text prompt. Therefore, your introduced cross-attention mechanism creates a new information pathway from the text prompt to the media representations. Is my understanding correct? And would you regard this as an important factor contributing to MokA's good performance?

---

> > > ### Author Response · Authors · 2025-08-03
> > >
> > > We appreciate the reviewer’s positive feedback on our rebuttal, and we will revise the relevant parts about “Multiple LoRA” to enhance clarity and readability.
> > >
> > > As per the follow-up question:
> > >
> > > Yes, cross-attention mechanism creates a new information pathway from the text prompt to the media representations. **It’s an important factor, but not the only one contributing to MokA's effectiveness. MokA’s effectiveness comes from two parts: the design of A/B matrices and cross-attention module.**
> > >
> > > 1. **MokA w/o cross-attention already brings noticeable performance improvements, since its design of A/B matrices is multimodal-aware.** As indicated in the following table, MokA w/o cross-attention has shown considerable improvement, since it can individually capture unimodal information by modality-specific matrices A, then modality-shared matrix B maps the separated unimodal subspaces back into a common latent space. The design of A/B matrices makes it better handle multimodal scenarios.
> > >
> > >     ---
> > >
> > >     |  | **MUSIC-AVQA** | **POPE** | **AIR-Bench** |
> > >     | --- | --- | --- | --- |
> > >     | LoRA | 73.41 | 70.28 | 31.75 |
> > >     | MokA w/o cross-attention | 74.85 | 73.57 | 33.25 |
> > >     | MokA | 75.71 | 74.23 | 39.64 |
> > >
> > >     ---
> > >
> > > 2. **Cross-attention module aims to enhance cross-modal interaction, and then helps to better capture the task-relevant information, further improving model performance.** As you mentioned, cross-attention module creates a new information pathway from the text prompt to the media representations. Then the interaction between the task (text token in the prompt) and media representations (non-text token in the prompt) is explicitly enhanced, thereby improving the capturing of the task-relevant information in the prompt and contributing to better performance. In addition, this cross-attention is only applied to the prompt part, hence it does not greatly break the causal nature when generating response. It targets to enhance the comprehension of prompt information.
> > > 3. **Since the cross-attention module primarily aims to explicitly strengthen cross-modal interactions (or task-media interaction), it is flexible and can be replaced by similar strategies.** We further explore self-attention and dynamic attention as alternative strategies. They can also explicitly achieve the interaction among tokens of different modalities. Experiments are conducted based on LLaMA2. The results indicate that these alternative mechanisms also obtain improvements, further demonstrating the flexibility of the cross-modal interaction module.
> > >
> > >     ---
> > >
> > >     |  | **MUSIC-AVQA** | **AVE** | **MME_${percep}$** | **MMBench** | **POPE** | **SEED-Bench** |
> > >     | --- | --- | --- | --- | --- | --- | --- |
> > >     | LoRA | 73.41 | 69.84 | 908.52 | 50.64 | 70.28 | 39.71 |
> > >     | MokA (cross-attention) | 75.71 | 74.68 | 1025.86 | 52.74 | 74.23 | 40.45 |
> > >     | MokA+self-attention | 75.13 | 74.92 | 1015.31 | 52.21 | 74.07 | 40.56 |
> > >     | MokA+dynamical attention | 74.97 | 74.48 | 1001.14 | 52.42 | 74.67 | 40.28 |
> > >
> > >     ---
> > >
> > >
> > > Overall, the design of A/B matrices and cross-attention module are both important to the effectiveness of MokA. The former helps to capture unimodal information individually, and the latter helps to the information interaction between the text prompt and the media representations.
> > >
> > > These discussions will be included in the revised manuscript. We thank you again for your confirmation of our rebuttal and your further discussion!

---

> ### Comment · Reviewer_x9zT · 2025-08-05
>
> Thanks for your response.
>
> After the discussion, this paper is complete and sound to me. I am therefore happy to raise my score.
>
> Meanwhile, since I have not been closely tracing the recent literature on this topic, for novelty and comparative experiments, I would suggest the AC to give more weight to the opinions of other reviews.

---

> > ### Author Response · Authors · 2025-08-05
> >
> > Thank you for these helpful suggestions, which improve the quality of the manuscript! We will include our discussion and additional experiments in the revised manuscript.

---

### Official Review · Reviewer_RCVF · 2025-07-03

**Clarity:** 3
**Significance:** 2
**Originality:** 3
**Rating:** 4
**Confidence:** 4

**Summary:**

This paper introduces MokA, a novel fine-tuning strategy for Multimodal Large Language Models (MLLMs) that addresses the limitations of directly applying LoRA—originally designed for LLMs—to MLLMs. The key insight is that shared LoRA parameters across modalities overly favor text, undermining the adaptation of non-text (e.g., visual, audio, speech) modalities. MokA uses modality-specific low-rank matrices (A) to capture unimodal features and introduces cross-modal attention between text and non-text tokens for explicit interaction, and projects all adapted features using a shared matrix (B) to unify representations.

**Questions:**

1. The paper should provide a comprehensive efficiency analysis of MokA, including computational overhead, training time, and memory requirements compared to baseline methods. Please elaborate on the trade-offs between performance and cost, and explain whether the performance gain comes from additional training, especially in resource-constrained environments.
2. The current formulation of "unimodal adaptation" in Equation 13 uses a shared B matrix across modalities, which contradicts the true definition of modality-specific adaptation. Additionally, the cross-modal interaction design omits Audio-Visual interactions. The authors should explain this design choice and provide ablation studies on whether incorporating Audio-Visual interactions would further enhance performance.
3. The authors should address how MokA would perform in scenarios with many modalities (3+) and propose potential solutions to mitigate the growing computational/inference burden in such cases.
4. The authors should explore and compare alternative attention mechanisms (such as bidirectional attention, dynamic attention weights, or gated attention) and explain why the current design was selected over potentially more powerful alternatives.

**Ethical Concerns:**

["NO or VERY MINOR ethics concerns only"]

**Final Justification:**

The authors have addressed most of my concerns in the rebuttal, and their clarifications have improved my understanding of the work. Therefore, I would like to raise my score to 4.

**Limitations:**

Yes

**Quality:**

3

**Strengths And Weaknesses:**

**Strength**：

1. The paper is well-written with clear motivation addressing the issue of text-dominance in LoRA-based MLLM adaptation. Additionally, the authors present the proposed algorithm MokA with clear illustrations and rigorous mathematical formulations, making it easy for readers to understand the methodology.
2. MokA is a logical extension of LoRA with minimal overhead, specifically adding modality-specific A matrices and a lightweight cross-attention module to effectively improve the performance of LoRA fine-tuning for MLLMs. While simple, the approach proves to be quite effective.
3. The extensive experimental results demonstrate that MokA outperforms multiple strong baselines across diverse benchmarks (MUSIC-AVQA, MMBench, POPE, AIR-Bench, etc.). Importantly, these performance gains are consistent across different LLM backbones.



**Weakness**：

1. In the experimental section, the paper primarily focuses on performance metrics while overlooking crucial efficiency indicators. The introduction of additional A matrices and the cross-attention module incurs extra computational and training time overhead during fine-tuning. Although MokA demonstrates performance gains, these improvements might simply result from increased training resources. The tradeoff between performance and cost is not deeply analyzed (e.g., FLOPs, wall-clock training time, memory usage).
2. Regarding Equation 13, the unimodal adaptation component is not entirely equivalent to "unimodal adaptation" since matrix B is shared across modalities rather than being modality-specific. Additionally, for the "cross-modal adaptation" component, we observe only Text-Visual and Text-Audio interactions, while Audio-Visual interactions are missing. The paper does not discuss whether introducing Audio-Visual interactions would further enhance model performance.
3. With each modality requiring its corresponding A matrix, the scalability of MokA becomes questionable when dealing with numerous modalities. This concern is particularly evident in Table 5, where even with just a few modalities, there's already a 1.13× increase in inference overhead. This overhead would grow substantially with additional modalities.
4. The design of the cross-attention module is fixed and simple. More flexible designs (e.g., bi-directional or dynamic attention) are not explored or compared.

---

> ### Author Rebuttal · Authors · 2025-07-31
>
> We sincerely thank the reviewer for these valuable suggestions, which have greatly strengthened our work. We will add these additional experiments and related discussion to the revised manuscript.
>
> ### **1. Efficiency evaluation & 3+ modality extension**
>
> We thank you for the valuable suggestions about efficiency evaluation. As suggested, we analyze and compare LoRA and MokA from computational cost, GPU memory usage, and extension under 3+ modality cases. Experiments are conducted based on LLaMA2.
>
> 1. **On Computational Cost:**
>
>     Compared to LoRA, MokA introduces additional A matrices and a cross-attention module. **However, it should be noted that the additional computational cost of MokA only comes from the cross-attention module, which is flexible and can be replaced by a more efficient strategy if needed.**
>
>     Concretely, in LoRA, each token is processed through *one* A matrix and *one* B matrix. Similarly, in MokA w/o cross-attention, each token passes through *one* modality-specific A matrix and *one* shared B matrix. This design maintains the same computational cost as LoRA while enabling modality awareness. In addition, as discussed in Sec. 4.4 and shown in the following table, **MokA w/o cross-attention already brings notable performance improvements over LoRA without additional computational burden.**
>
>     ---
>
>     |  | **MUSIC-AVQA** | **POPE** | **AIR-Bench** |
>     | --- | --- | --- | --- |
>     | LoRA | 73.41 | 70.28 | 31.75 |
>     | MokA w/o cross-attention | 74.85 | 73.57 | 33.25 |
>     | MokA | 75.71 | 74.23 | 39.64 |
>
>     ---
>
>     As per the cross-attention module, it is employed to explicitly strengthen the interaction between text and non-text tokens, thereby facilitating cross-modal adaptation. **Alternative strategies, like more efficient cross-modal interaction methods, can also be considered** as mentioned in the manuscript. Here we provide one variant, MokA w/ naive interaction, which performs a simple, uniform mapping from text tokens to non-text tokens without employing any attention mechanism. And this more efficient variant also achieves improvement.
>
>     ---
>
>     |  | **MUSIC-AVQA** | **AVE** | **MME_${percep}$** | **MMBench** | **POPE** | **SEED-Bench** |
>     | --- | --- | --- | --- | --- | --- | --- |
>     | LoRA | 73.41 | 69.84 | 908.52 | 50.64 | 70.28 | 39.71 |
>     | MokA | 75.71 | 74.68 | 1025.86 | 52.74 | 74.23 | 40.45 |
>     | MokA w/ naive interaction | 75.04 | 73.18 | 996.73 | 51.49 | 73.52 | 40.17 |
>
>     ---
>
> 2. **On GPU Memory Usage:**
>
>     MokA’s additional GPU memory usage mainly comes from having modality-specific A matrices. In practice, the A matrices are lightweight by design (following LoRA’s low-rank parameterization), so the increase of memory usage is small and remains within acceptable limits. We will provide more information in the following parts.
>
> 3. **3+ Modality Extension:**
>
>     Based on the suggestion, we extend our experiments to scenarios with more than three modalities. Specifically, we consider a **four-modality setting** involving audio, visual, point cloud, and language data, and evaluate it on the MCUB-3 benchmark. In this 3+ modality case, LoRA has an accuracy of 37.41%. Our MokA has an accuracy of 45.58%, indicating its scalability and effectiveness under 3+ modality cases. We also provide the efficiency comparison in the following parts.
>
> 4. **On Comprehensive Efficiency Analysis:**
>
>     Following the reviewer’s valuable suggestion, we have added a more detailed efficiency analysis comparing **FLOPs, GPU memory usage**, and **average forward time per sample** (proportional to training time). For clarity, we report these metrics for the two-modality (VL), three-modality (AVL), and newly added four-modality (AVPL) settings. As shown in the tables, MokA’s extra computational and memory cost remains limited and acceptable for typical multimodal scenarios, which aligns with the above analysis. **Overall, we obtain significant performance gains with only modest extra resource consumption.**
>
>     ---
>
>     | **VL (MME_${percep}$)** | **FLOPs** | **Memory Usage** | **Avg. forward time/sample** |
>     | --- | --- | --- | --- |
>     | LoRA | 1526.25 GFLOPs (1.000x) | 27315.75 MB  (1.000x) | 1.75 s (1.000x) |
>     | MokA | 1539.55 GFLOPs (1.009x) | 27329.78 MB (1.001x) | 1.87 s (1.069x) |
>
>     ---
>
>     | **AVL (MUSIC-AVQA)** | **FLOPs** | **Memory Usage** | **Avg. forward time/sample** |
>     | --- | --- | --- | --- |
>     | LoRA | 12116.72 GFLOPs  (1.000x) | 27961.74 MB (1.000x) | 5.53 s (1.000x) |
>     | MokA | 12375.90 GFLOPs (1.021x) | 27996.49 MB (1.001x) | 6.27 s  (1.134x) |
>
>     ---
>
>     | **AVPL (MCUB-3)** | **FLOPs** | **Memory Usage** | **Avg. forward time/sample** |
>     | --- | --- | --- | --- |
>     | LoRA | 12620.51 GFLOPs  (1.000x) | 28329.56 MB (1.000x) | 1.64 s (1.000x) |
>     | MokA | 12778.88 GFLOPs (1.013x) | 28377.37 MB (1.002x) | 1.99 s  (1.213x) |
>
>     ---
>
>
> ---
>
> ### **2. Uni-modal adaptation in Eq. (13)**
>
> Specifically, in Eq. (13), the unimodal adaptation component is formulated as $\[BA^a\mathbf{x}^a; BA^v\mathbf{x}^v; BA^t\mathbf{x}^t\]$, where $A^a$, $A^v$, $A^t$ are modality-specific matrices and $B$ is a shared projection matrix. Input of modality $i$ first undergoes a unique modality-specific transformation through its respective $A^i$ matrix. And then, the subsequent shared $B$ matrix applies an **identical mapping** to all modalities. Therefore, **the modality-specific characteristics introduced by $A^i$ and $\mathbf{x}^i$ are preserved, since $B$ is identical to all modality-specific terms and it does not explicitly mix information across modalities.**
>
> Therefore, we consider this component an **approximation of unimodal adaptation**, since the distinct modality-specific paths remain independent. In the revised manuscript, we will refine the related analysis.
>
> ---
>
> ### **3. Audio-visual interaction**
>
> In the original MokA, only the attention between text and non-text tokens is considered. It is motivated by the fact that text modality typically conveys the question or task description, while the audio and visual modalities provide environment information, in the instruction. Therefore, cross-attention is applied to explicitly enhance the interaction between the task (text token) and environment (non-text token).
>
> As suggested, it is also worth exploring whether further interactions between the scene modalities themselves—i.e., audio and visual—can be beneficial. To this end, we conduct additional ablation studies. Experiments are conducted based on LLaMA2. The table reports the results for audio-visual attention with both audio as query and video as query. The results show that introducing additional audio-visual attention can bring gains but is not very noticeable.  The reason could be that the original cross-attention in MokA has effectively captured task-relevant information within the audio and visual inputs. As a result, the potential benefit from further enhancing explicit audio-visual interactions is relatively limited. We will include this new ablation and the corresponding discussion in the revised manuscript.
>
> ---
>
> |  | **MUSIC-AVQA** | **AVE** |
> | --- | --- | --- |
> | LoRA | 73.41 | 69.84 |
> | MokA | 75.71 | 74.68 |
> | MokA w/ audio query att. | 75.78 | 74.53 |
> | MokA w/video query att. | 75.76 | 74.81 |
>
> ---
>
> ### **4. More attention strategies**
>
> We thank the reviewer for this valuable suggestion regarding the flexibility of the cross-attention module. **As mentioned above, our cross-attention design primarily aims to explicitly strengthen cross-modal interactions, and it is flexible and can be replaced by different strategies if needed.** Based on the suggestion, we further explore bi-directional attention and dynamic attention as alternative designs. Experiments are conducted based on LLaMA2.
>
> The results show that these alternative designs also achieve improvements over the LoRA baseline, further demonstrating the flexibility and effectiveness of the cross-modal interaction module. We appreciate the reviewer’s suggestion, which has strengthened the paper by motivating us to include these informative comparisons.
>
> ---
>
> |  | **MUSIC-AVQA** | **AVE** | **MME_${percep}$** | **MMBench** | **POPE** | **SEED-Bench** |
> | --- | --- | --- | --- | --- | --- | --- |
> | LoRA | 73.41 | 69.84 | 908.52 | 50.64 | 70.28 | 39.71 |
> | MokA | 75.71 | 74.68 | 1025.86 | 52.74 | 74.23 | 40.45 |
> | bi-att. | 75.13 | 74.92 | 1015.31 | 52.21 | 74.07 | 40.56 |
> | dynamical att. | 74.97 | 74.48 | 1001.14 | 52.42 | 74.67 | 40.28 |
>
> ---

---

> ### Author Response · Authors · 2025-08-04
>
> Dear reviewer, we would like to know if our responses have successfully tackled your concerns. We are open to answering any additional questions that you may have.

---

> > ### Comment · Reviewer_RCVF · 2025-08-07
> >
> > Thanks for the rebuttal provided by the authors. The responses do solve most of my concerns and successfully clarify some details in the paper. I suggest add part of the rebuttal to the revised paper. I would like to raise my score to 4.

---

> > > ### Author Response · Authors · 2025-08-07
> > >
> > > Thanks to the reviewers' insightful and valuable suggestions, which definitely help us improve this paper! We will include newly added experiments and discussion in the revised version.

---

### Note · Authors · 2025-08-15

We are deeply grateful to the reviewers and the AC for their time, insightful feedback, and constructive suggestions that significantly strengthened our manuscript.

In this work, we reveal that most current efficient multimodal fine-tuning methods are directly borrowed from LLMs, often neglecting the intrinsic differences of multimodal scenarios and hindering full utilization of all modalities. Therefore, we introduce Multimodal low-rank Adaptation (MokA), a multimodal-aware fine-tuning strategy that compresses unimodal information via modality-specific parameters while explicitly enhancing cross-modal interaction. Extensive experiments cover multiple representative multimodal scenarios and LLM backbones.

We thank the reviewers’ positive comments on our work (*e.g., logical extension of LoRA with minimal overhead, clear logical flow, clear motivation, comprehensive experiments, well-written*). More importantly, we also sincerely thank all the reviewers’ valuable suggestions, including more detailed efficiency analysis, scalability, rank selection, related work discussion, and clarification of technical details.

We are happy that all the reviewers’ concerns were successfully addressed during the rebuttal stage. The newly added experiments and discussions will be incorporated into the revised version to further enhance the manuscript. We once again appreciate the reviewers and the AC for their insightful feedback and constructive guidance.

---

### Decision · Program_Chairs · 2025-09-17

**Decision:**

Accept (oral)

**Comment:**

Lora fine-tuning, when applied directly to MLLMs, overly favors text, and undermines the adaptation of non text modalities. To address this, the authors present Moka, which uses modality specific low rank matrices for unimodal features, and a cross-modal attention to model the interaction explicitly, and finally, projects all features to a shared space for unification. This outperforms multiple strong baselines across diverse benchmarks (MUSIC-AVQA, MMBench, POPE, AIR-Bench, etc.). Importantly, these performance gains are consistent across different LLM backbones.

Strengths

- All reviewers find the paper very well motivated, clearly illustrated with a clear logical flow and excellent figures, and mathematically rigorous, making the paper easy to read and understand.
- The approach is simple i.e. only adds a modality specific module and a cross-attention module, but yet achieves very strong results.
- All reviewers appreciate the strong experimental design and the strong results on multiple benchmarks.

Weaknesses

1. The primary weaknesses/questions by reviewers have all been promptly and very reasonably addressed by the authors. These include:
    1. Full absolute FLOPs, memory, and latency numbers on both Lora and Moka.
    2. Adding audio-visual interaction in addition to text-audio and text-visual interactions.
    3. Stronger versions of the cross-attention module
    4. Ablations of higher/lower ranks for Lora - doesn’t help Lora
    5. Scaling to 3+ modalities - Moka scales well
    6. Comparison with Omni-SMoLA - Moka matches it with a much simpler and more lightweight design.
    7. Catastrophic forgetting - On the MME reasoning benchmark, the LoRA baseline achieves an accuracy of 16.78%, while MokA reaches 18.01%, indicating that MokA has stronger capacity in this aspect.

The paper has all four accepts, including a strong accept. It is a solidly executed paper with a clear motivation and a strong/easily readable manuscript. Based on results, this paper can serve as the de-facto PEFT method for MLLMs going forward. I recommend this be accepted.